# Indocyanine green fluorescence imaging-guided versus conventional laparoscopic lymphadenectomy for gastric cancer: long-term outcomes of a phase 3 randomised clinical trial

Indocyanine green (ICG) fluorescence imaging-guided lymphadenectomy has been demonstrated to be effective in increasing the number of lymph nodes (LNs) retrieved in laparoscopic gastrectomy for gastric cancer (GC). Previously, we reported the primary outcomes and short-term secondary outcomes of a phase 3, open-label, randomized clinical trial (NCT03050879) investigating the use of ICG for image-guided lymphadenectomy in patients with potentially resectable GC. Patients were randomly (1:1 ratio) assigned to either the ICG or non-ICG group. The primary outcome was the number of LNs retrieved and has been reported. Here, we report the primary outcome and long-term secondary outcomes including three-year overall survival (OS), three-year disease-free survival (DFS), and recurrence patterns. The per-protocol analysis set population is used for all analyses (258 patients, ICG [n = 129] vs. non-ICG group [n = 129]). The mean total LNs retrieved in the ICG group significantly exceeds that in the non-ICG group ($50.5 \pm 15.9$ vs $42.0 \pm 10.3$, $P < 0.001$). Both OS and DFS in the ICG group are significantly better than that in the non-ICG group (log-rank $P = 0.015$; log-rank $P = 0.012$, respectively). There is a difference in the overall recurrence rates between the ICG and non-ICG groups (17.8% vs 31.0%). Compared with conventional lymphadenectomy, ICG guided laparoscopic lymphadenectomy is safe and effective in prolonging survival among patients with resectable GC.

Gastric cancer (GC) accounts for approximately 8% of cancer-related deaths worldwide[1]. It is necessary to focus on treatment strategies to reduce cancer-related mortality and improve survival. Lymph node (LN) metastasis is closely related to tumor staging and postoperative adjuvant therapy. Because of the high metastasis rate of perigastric LNs, radical lymphadenectomy of metastatic LNs is the cornerstone of surgical treatment of GC[2,3].

Nonetheless, lymphadenectomy is often merely performed based on the surgeon's preference and experience. However, owing to the complex vascular anatomy and lymphatic drainage around the stomach, efficient and accurate dissection of LNs without increasing surgery-related complications remains a substantial challenge for surgeons, especially junior trained ones. Recently, with the successful application of indocyanine green (ICG) fluorescence imaging

✉ e-mail: wwkzch@163.com; xjwhw2019@163.com; hcmlr2002@163.com

technology in minimally invasive surgery[4], surgeons have found that ICG fluorescence imaging has good tissue penetration and can identify LNs in hypertrophic adipose tissue better than other dyes. Minimally invasive lymphadenectomy for GC guided by ICG fluorescence imaging has become a new exploration direction for individualized and precise treatment[5,6].

Retrospective studies[7–9] have shown that ICG can achieve good mapping of perigastric LNs, thus significantly increasing the total number of LNs retrieved during laparoscopic gastrectomy. However, the lack of high-quality evidence impedes scientific conclusions regarding the oncological efficacy of ICG fluorescence imaging-guided lymphadenectomy. Therefore, the Fujian Medical University Union Hospital Gastric Surgery Study Group (FUGES) conducted a randomized controlled trial (RCT) to evaluate the efficacy of ICG fluorescence imaging-guided laparoscopic lymphadenectomy for GC. Previously, we reported that ICG imaging effectively improved the number of LNs retrieved and reduced LN dissection noncompliance without increasing complications in patients undergoing D2 lymphadenectomy[10]. However, the long-term oncological efficacy of ICG tracer-guided lymphadenectomy in GC patients remains unclear. Long-term oncological efficacy is an important basis for determining whether a new technology or method can be applied in clinical practice.

In this work, we present the subsequent follow-up results of FUGES-012, wherein the long-term oncologic outcomes of 3-year overall survival (OS), 3-year disease-free survival (DFS), and recurrence patterns are reported.

## Results
### Baseline characteristics
From November 19, 2018 to July 13, 2019, 266 patients were randomly assigned to the ICG and non-ICG groups (n = 133 per group). Four patients were excluded from the ICG group: one with ICG contamination, one with an unresectable tumor, one with peritoneal metastasis, and one who withdrew from the study. Meanwhile, four patients were excluded from the non-ICG group: one with an unresectable tumor, two with peritoneal metastases, and one who withdrew from the study[10]. Finally, 129 patients (86 men and 43 women) in the ICG group and 129 patients in the non-ICG group (87 men and 42 women) were included in the per-protocol analysis (Fig. 1).

The baseline and postoperative patient characteristics are shown in Table 1. The baseline characteristics were mostly balanced, except for the distribution of tumor locations across the groups. The mean number of LNs retrieved in the ICG group was significantly higher than that in the non-ICG group (mean [SD], 50.5 [15.9] vs. 42.0 [10.3], respectively; P < 0.001)[10]. The rates of adjuvant chemotherapy were comparable between the two groups (51.9% vs. 59.7%; P = 0.210). Upon further subgroup analysis (Supplementary Information 1: Supplementary Table 1) showed that regardless of tumor location, the number of LNs retrieved in the ICG group was higher than that of the non-ICG group (Lower: mean [SD], 50.0 [16.6] vs. 40.8 [9.9], P = 0.001; Upper/Middle: mean [SD], 51.1 [14.9] vs. 42.8 [10.4]). No significant difference in LN dissection noncompliance and compliance between different BMI categories in the ICG group patients (P = 0.627) (Supplementary Table 2). There were no significant between-group differences in the chemotherapy regimen type, completion rate of adjuvant chemotherapy, or time to adjuvant chemotherapy initiation (Supplementary Table 3). The median follow-up period was 40.0 months (interquartile range, 38.0–41.0 months).

The ITT analysis was performed excluding those who withdrew consent preoperatively or who had unresectable GC detected intraoperatively. In the intention-to-treat (ITT) analysis, there were 130 patients in the ICG group and 129 patients in the non-ICG group, with a mean (SD) total number of LNs retrieved of 50.6 (15.9) and 42.0 (10.3),

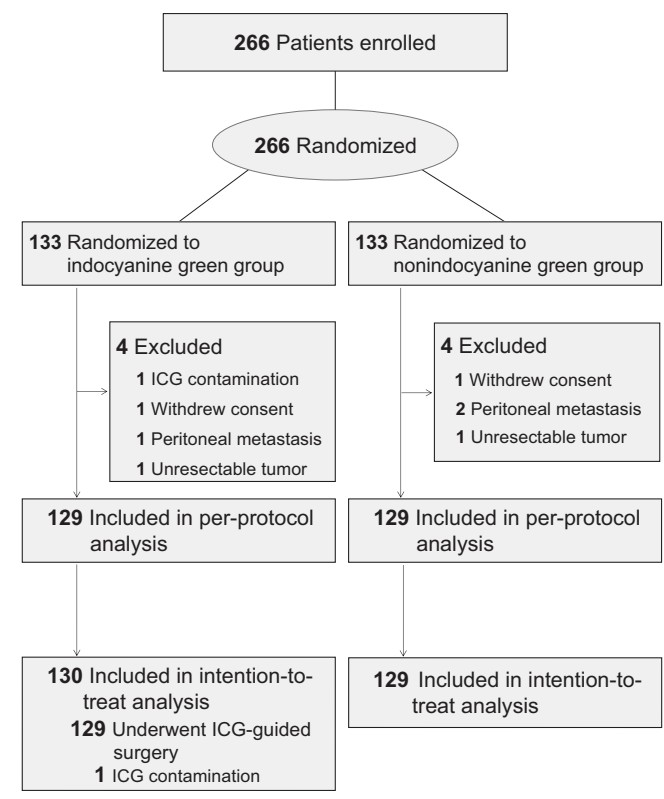

**Fig. 1 | Trial Profile.**

respectively (P < 0.001). The median follow-up period was 40.0 months (interquartile range, 38.0–41.0 months).

### Overall survival
The deaths of 52 patients resulted in a 3-year actual OS rate of 86.0% (18 of 129) in the ICG group and 73.6% (34 of 129) in the non-ICG group. The OS in the ICG group was statistically significantly better than that in the non-ICG group (log-rank P = 0.015), wherein the 3-year OS rate in the ICG group and non-ICG group was 86.0% (95%CI = 80.1%–92.0%) and 73.6% (95%CI = 66.0%–81.2%) (Fig. 2); There was a significant difference (Table 2) in the cumulative incidence for cancer-specific death between the two groups (ICG vs non-ICG = 13.2% vs. 24.8%, risk difference = −0.116, HR = 0.53; 95%CI, 0.29-0.96; adjusted P = 0.037). Multivariable Cox regression analysis (Table 3) revealed that ICG fluorescence imaging-guided lymphadenectomy was an independent protective factor for OS compared with the non-ICG group after adjusting for age, tumor location, lymphovascular invasion, tumor size, pathological stage, and adjuvant chemotherapy (HR = 0.49; 95% CI, 0.27–0.91; P = 0.023).

In the ITT analysis, the survival analysis revealed that the OS of the ICG group was superior to that of the non-ICG group (log-rank P = 0.014) (Supplementary Fig. 1).

### Disease-free survival
The DFS in the ICG group was statistically significantly better than that in the non-ICG group (log-rank P = 0.012). The 3-year actual DFS rates were 81.4% (95%CI = 74.7%–88.1%, 105 of 129) in the ICG group and 68.2% (95%CI = 58.3%–75.1%, 88 of 129) in the non-ICG group, with an absolute difference of 13.2% (Fig. 2). Univariable Cox regression analysis (Table 3) showed that compared with conventional lymphadenectomy, ICG fluorescence imaging-guided lymphadenectomy was associated with better DFS in patients (HR = 0.53; 95%CI, 0.32-0.88; P = 0.014). A similar HR (multivariable Cox regression analysis) was observed after adjusting for tumor location, lymphovascular invasion,

## Table 1 | Baseline and Postoperative Characteristics of the ICG Group and Non-ICG Group

| Characteristic | No. (%) / Mean (SD) | | P Value |
|---|---|---|---|
| | ICG (n = 129) | Non-ICG (n = 129) | |
| Age, years | 57.8 (10.7) | 60.1 (9.1) | 0.071 |
| BMI, kg/m² | 23.2 (3.2) | 22.8 (3.1) | 0.263 |
| Sex | | | |
| Male | 86 (66.7) | 87 (67.4) | 0.895 |
| Female | 43 (33.3) | 42 (32.6) | |
| ECOG PS | | | |
| 0 | 114 (88.4) | 113 (87.6) | 0.848 |
| 1 | 15 (11.6) | 16 (12.4) | |
| Tumor location | | | |
| Upper | 33 (25.6) | 66 (51.2) | <0.001 |
| Middle | 21 (16.3) | 14 (10.9) | |
| Lower | 75 (58.1) | 49 (38.0) | |
| Lymphvascular invasion | | | |
| Negative | 66 (51.2) | 73 (56.6) | 0.382 |
| Positive | 63 (48.8) | 56 (43.4) | |
| Size, cm | | | |
| ≤3 | 64 (49.6) | 50 (38.8) | 0.079 |
| >3 | 65 (50.4) | 79 (61.2) | |
| cT stage | | | |
| cT1 | 35 (27.1) | 29 (22.5) | 0.645 |
| cT2-cT3 | 68 (52.7) | 70 (54.3) | |
| cT4a | 26 (20.2) | 30 (23.3) | |
| cN stage | | | |
| cN0 | 60 (46.5) | 54 (41.9) | 0.452 |
| cN+ | 69 (53.5) | 75 (58.1) | |
| pT stage | | | |
| pT1 | 42 (32.6) | 39 (30.2) | 0.687 |
| pT2-T4a | 87 (67.4) | 90 (69.8) | |
| pN stage | | | |
| pN0 | 54 (41.9) | 55 (42.6) | 0.414 |
| pN1 | 24 (18.6) | 16 (12.4) | |
| pN2 | 20 (15.5) | 18 (14.0) | |
| pN3 | 31 (24.0) | 40 (31.0) | |
| AJCC7th staging | | | |
| I | 50 (38.8) | 41 (31.8) | 0.429 |
| II | 33 (25.6) | 33 (25.6) | |
| III | 46 (35.7) | 55 (42.6) | |
| Metastatic LNs | 5.6 (11.2) | 5.7 (8.9) | 0.941 |
| Total LN retrieved | 50.5 (15.9) | 42.0 (10.3) | <0.001 |
| ≥30 | 129 (100.0) | 113 (87.6) | <0.001 |
| <30 | 0 (0.0) | 16 (12.4) | |
| LNs dissection compliance | | | |
| Noncompliance | 41 (31.8) | 74 (57.4) | <0.001 |
| Compliance | 88 (68.2) | 55 (42.6) | |
| Postoperative complication | | | |
| No | 109 (84.5) | 108 (83.7) | 0.863 |
| Yes | 20 (15.5) | 21 (16.3) | |
| Received adjuvant chemotherapy | | | |
| No | 62 (48.1) | 52 (40.3) | 0.210 |
| Yes | 67 (51.9) | 77 (59.7) | |

*AJCC* American Joint Committee on Cancer, *BMI* body mass index (calculated as weight in kilograms divided by height in meters squared), *ECOG PS* Eastern Cooperative Oncology performance status, *ICG* indocyanine green, *LN* lymph node, *cT* clinical T, *cN* clinical N, *pT* pathological T, *pN* pathological N.

tumor size, pathological stage, and adjuvant chemotherapy (HR = 0.51; 95%CI, 0.30-0.87; P = 0.014).

In the ITT analysis, the survival analysis revealed that the DFS of the ICG group was better than that of the non-ICG group (log-rank P = 0.011) (Supplementary Fig. 1).

### Recurrence

Within the first three years of follow-up, recurrence was found in 23 (cumulative incidence, 17.8%) and 40 (cumulative incidence, 31.0%) patients in the ICG and non-ICG groups, respectively (Table 2). The cumulative incidence of recurrence is shown in Supplementary Fig. 2.

Treating death as the competing risk, a significant difference in the cumulative recurrence incidence was found between the ICG and non-ICG groups (HR = 0.54; 95% CI, 0.32–0.91; adjusted P = 0.020). The risk difference was −0.131. The cumulative incidence of locoregional recurrence was significantly different between the two groups (ICG vs. non-ICG, 1.6% vs. 7.8%, HR = 0.22; 95% CI, 0.05-0.99; adjusted P = 0.048). The risk difference was −0.073. However, the cumulative incidence of recurrence in the peritoneum, liver, multiple sites, and other sites did not significantly differ between the two groups (all P > 0.05).

### Incremental harvested lymph nodes in ICG group improve survival

For pN0 patients, there is no statistically significant difference in prognosis between ICG and non-ICG patients (OS: P = 0.083; DFS: P = 0.083). However, for pN+ patients, the prognosis of ICG patients is significantly better than that of non-ICG patients (OS: P = 0.023; DFS: P = 0.012) (Supplementary Fig. 3). In the ICG group, the number of LNs retrieved in each patient was ≥30; in the non-ICG group, 16 patients (12.4%) had <30 retrieved LNs (P < 0.001). Supplementary Fig. 4 shows that the OS of patients with <30 retrieved LNs was significantly lower than that of patients with ≥30 retrieved LNs (log-rank P = 0.030). The 3-year OS rate of patients with <30 retrieved LNs was 56.3%, which was significantly lower than the 81.4% of patients with ≥30 retrieved LNs. The DFS of patients with <30 retrieved LNs was comparable to that of patients with ≥30 retrieved LNs (log-rank P = 0.148). The 3-year DFS rate of patients with <30 retrieved LNs was 56.3%, which was also lower than the 76.0% of patients with ≥30 retrieved LNs, although the difference was not statistically significant.

Further analysis of patients with ≥30 retrieved LNs revealed that the OS in the ICG group was significantly higher than that in the non-ICG group (log-rank P = 0.047; 3-year OS rate: 86.0% vs. 76.1%) (Supplementary Fig. 5). The 3-year recurrence rate in the ICG group was significantly lower than that in the non-ICG group (17.8% [23/129] vs. 30.1% [34/113]; χ², P = 0.025). The DFS in the ICG group was also significantly higher than that in the non-ICG group (log-rank P = 0.026; 3-year DFS rate: 81.4% vs. 69.9%).

Multivariable Cox regression analysis (Supplementary Table 4) showed that compared with conventional lymphadenectomy, ICG fluorescence imaging-guided lymphadenectomy was an independent protective factor for DFS in patients with ≥30 retrieved LNs (ICG vs. non-ICG, HR = 0.52; 95%CI, 0.29–0.92; P = 0.024). While univariable and multivariable Cox regression analysis indicated that ICG fluorescence imaging-guided lymphadenectomy was not an independent factor for OS in patients with ≥30 retrieved LNs (P > 0.05).

### ICG reduces the LN dissection noncompliance and locoregional recurrence, and improves DFS

The LN dissection noncompliance rate in the ICG group (41 of 129 patients [31.8%]) was lower than that in the non-ICG group (74 of 129 patients [57.4%]; P < 0.001). In the overall population, the OS (log-rank P = 0.412; 3-year OS rates: 81.8% vs 77.4%) and DFS (log-rank P = 0.238; 3-year DFS rates: 77.6% vs 71.3%) of patients with compliant and noncompliant lymphadenectomy was comparable (Supplementary Fig. 6). Supplementary Table 5 lists the clinical characteristics of each patient

who experienced locoregional recurrence within 3 years. The LN dissection noncompliance rate in patients with locoregional recurrence was significantly higher (75.0%, 9/12) than that in patients without locoregional recurrence (43.1%, 106/246; P = 0.030). In addition, 50% of patients (1/2) in the ICG group underwent noncompliant lymphadenectomy, while 80% of patients (8/10) in the non-ICG group underwent noncompliant lymphadenectomy.

The OS of pN0 patients with compliant and noncompliant lymphadenectomy were comparable (log-rank P = 0.578; 3-year OS rates: 98.1% vs. 96.4%), and the 3-year OS rate of pN+ patients with compliant lymphadenectomy was 71.9%, which was better than the 60.0% of pN+ patients with noncompliant lymphadenectomy; although the difference was not statistically significant (log-rank P = 0.151, Supplementary Fig. 7). The DFS of pN0 patients with compliant and noncompliant lymphadenectomy were comparable (log-rank P = 0.578; 3-year DFS rates: 98.1% vs. 96.4%), whereas the DFS of pN+ patients with compliant

lymphadenectomy was significantly better than that of pN+ patients who underwent noncompliant lymphadenectomy (log-rank P = 0.043), the 3-year DFS rate of pN+ patients with compliant lymphadenectomy was 65.2%, which was better than the 48.3% of pN+ patients who underwent noncompliant lymphadenectomy. The 3-year cumulative recurrence rate of pN+ patients who underwent noncompliant lymphadenectomy was 50% (30/60), which was higher than the 34.8% (31/89) of pN+ patients who underwent compliant lymphadenectomy ($\chi^2$ = 0.065).

Further analysis (Supplementary Fig. 8) showed that the OS of pN0 patients in the ICG and non-ICG groups were comparable (log-rank P = 0.083; 3-year OS rates: 100.0% vs. 94.5%), and the OS of pN+ patients in the ICG group was significantly better than that of pN+ patients in the non-ICG group (log-rank P = 0.023; 3-year OS rate: 76.0% vs 58.1%). The DFS of pN0 patients in the ICG and non-ICG groups were comparable (log-rank P = 0.083; 3-year DFS rates: 100.0% vs. 94.5%), and the DFS of pN+ patients in the ICG group was

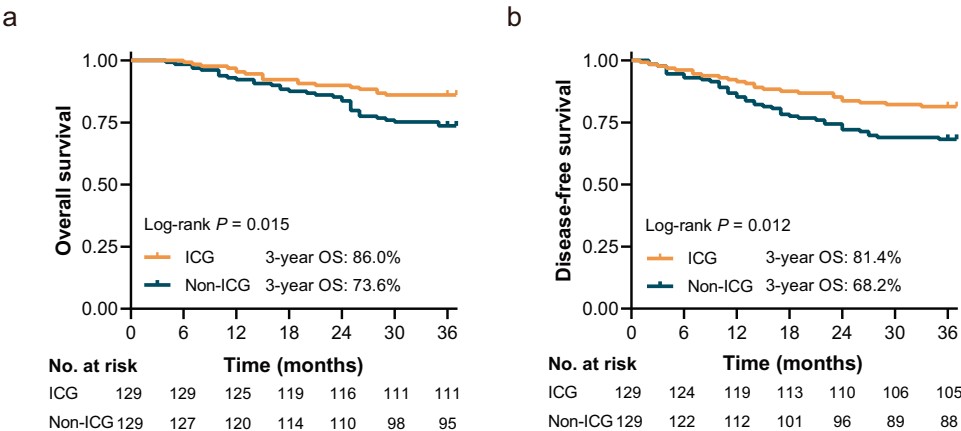

**Fig. 2 | Survival Analysis of the ICG Group and Non-ICG Group. a** Comparing Overall Survival Between the ICG Group and Non-ICG Group. **b** Comparing Disease-free Survival Between the ICG Group and Non-ICG Group.

**Table 2 | Frequencies of Causes of First Recurrence and Death Within 3 Years After Surgery in ICG and Non-ICG Groups**

| Events | Surgery, No. (%) | | Risk Difference[a] | Hazard Ratio (95% CI)[b] | P Value[c] | Hazard Ratio (95% CI)[d] | Adjusted P Value[e] |
|---|---|---|---|---|---|---|---|
| | ICG Group (n = 129) | Non-ICG Group (n = 129) | | | | | |
| Recurrence[f] | 23 (17.8) | 40 (31.0) | −0.131 | 0.53 (0.32–0.89) | 0.017 | 0.54 (0.32–0.91) | 0.020 |
| Local | 2 (1.6) | 10 (7.8) | −0.073 | 0.19 (0.04–0.85) | 0.030 | 0.22 (0.05–0.99) | 0.048 |
| Peritoneum | 9 (7.0) | 10 (7.8) | −0.009 | 0.87 (0.35–2.13) | 0.752 | 0.96 (0.39–2.36) | 0.923 |
| Liver | 2 (1.6) | 6 (4.7) | −0.032 | 0.33 (0.07–1.62) | 0.170 | 0.31 (0.06–1.56) | 0.155 |
| Multiple sites[g] | 4 (3.1) | 5(3.9) | −0.009 | 0.79 (0.21–2.92) | 0.718 | 0.93 (0.25–3.51) | 0.917 |
| Other or uncertain sites[h] | 6 (4.7) | 9 (7.0) | −0.026 | 0.64 (0.23–1.81) | 0.402 | 0.55 (0.19–1.60) | 0.274 |
| All-cause death[i] | 18 (14.0) | 34 (26.4) | −0.124 | 0.50 (0.28–0.89) | 0.018 | 0.54 (0.30–0.96) | 0.035 |
| Gastric cancer | 17 (94.4) | 32 (94.1) | −0.116 | 0.50 (0.28–0.91) | 0.022 | 0.53 (0.29–0.96) | 0.037 |
| Other causes[j] | 1 (5.6) | 2 (5.9) | −0.011 | 0.47 (0.04–5.21) | 0.540 | 0.48 (0.04–5.43) | 0.556 |

[a]For recurrence, the risk difference was calculated by subtracting the cumulative incidence in the first 3 years of the Non-ICG group from that of the ICG group, in the presence of competing events; for all-cause death, the risk difference was calculated by subtracting the 3-year overall survival rate of the Non-ICG group from that of the ICG group.
[b]For recurrence, competing-risks survival regression was used to derive the hazard ratio, 95% CI, and P value. For total recurrence, all-cause death was the competing event; for the specific types of recurrence, other types of recurrence and death were the competing events; for gastric cancer cause of death, other causes of death were the competing events, and vice versa. Univariable Cox regression was used for recurrence and all-cause death. Non-ICG group is the reference group.
[c]P value for the hazard ratios.
[d]Multivariable Cox regression was used for recurrence and all-cause death, adjustment for sex, AJCC7th staging, and adjuvant chemotherapy.
[e]Adjusted P value for the hazard ratios.
[f]Refers only to first-time recurrence, even though patients can have recurrence at multiple times.
[g]Includes patients who have recurrence simultaneously in 2 or more metastatic sites, including peritoneum, liver, lung, bone, brain, distant lymph node, or other hematogenous metastatic sites.
[h]Includes hematogenous recurrence at sites other than the liver (ie, lung, bone, brain, adrenal gland), recurrence at distant lymph node, and recurrence at uncertain sites.
[i]Post hoc exploratory outcomes. All-cause death includes death from gastric cancer and other causes.
[j]Includes other cancers, diseases other than cancer, unintentional injuries, and unknown causes.

**Table 3 | Univariable and Multivariable Cox Regression Analyses of Risk Factors for Survival**

| Clinicopathologic Parameters | Overall Survival | | | | Disease-free Survival | | | |
|---|---|---|---|---|---|---|---|---|
| | Univariable Model | | Multivariable Model | | Univariable Model | | Multivariable Model | |
| | HR (95%CI) | P | HR (95%CI) | P | HR (95%CI) | P | HR (95%CI) | P |
| Group | | | | | | | | |
| Non-ICG | Ref | | Ref | | Ref | | Ref | |
| ICG | 0.50 (0.28–0.89) | 0.018 | 0.49 (0.27-0.91) | 0.023 | 0.53 (0.32-0.88) | 0.014 | 0.51 (0.30-0.87) | 0.014 |
| Age, year | | | | | | | | |
| ≤60 | Ref | | Ref | | Ref | | | |
| >60 | 2.09 (1.17–3.74) | 0.012 | 1.87 (0.99-3.52) | 0.053 | 1.64 (1.00-2.70) | 0.050 | | |
| Sex | | | | | | | | |
| Female | Ref | | | | Ref | | | |
| Male | 1.31 (0.71–2.43) | 0.383 | | | 1.05 (0.63-1.77) | 0.847 | | |
| BMI, kg/m² | | | | | | | | |
| <25 | Ref | | | | Ref | | | |
| ≥25 | 0.72 (0.37–1.40) | 0.329 | | | 0.65 (0.35–1.19) | 0.161 | | |
| ECOG PS | | | | | | | | |
| 0 | Ref | | | | Ref | | | |
| 1 | 1.42 (0.67–3.02) | 0.360 | | | 1.39 (0.71–2.72) | 0.338 | | |
| Tumor location | | | | | | | | |
| Lower | Ref | | Ref | | Ref | | Ref | |
| Middle | 2.23 (0.99–5.06) | 0.054 | 1.95 (0.83-4.58) | 0.125 | 2.34 (1.14–4.79) | 0.020 | 1.80 (0.85-3.80) | 0.124 |
| Upper | 2.25 (1.21–4.18) | 0.010 | 1.00 (0.52–1.93) | 0.997 | 2.31 (1.33–4.02) | 0.003 | 1.16 (0.65-2.06) | 0.616 |
| Histology | | | | | | | | |
| Differentiated | Ref | | | | Ref | | | |
| Undifferentiated | 1.47 (0.85–2.54) | 0.170 | | | 1.36 (0.83–2.21) | 0.221 | | |
| Lymphvascular invasion | | | | | | | | |
| Negative | Ref | | Ref | | Ref | | Ref | |
| Positive | 3.30 (1.81–6.02) | <0.001 | 1.22 (0.61–2.46) | 0.571 | 4.55 (2.59–8.00) | <0.001 | 1.62 (0.84-3.11) | 0.147 |
| Size, cm | | | | | | | | |
| ≤3 | Ref | | Ref | | Ref | | Ref | |
| >3 | 2.69 (1.43–5.03) | 0.002 | 1.02 (0.53–1.97) | 0.954 | 3.81 (2.08–6.99) | <0.001 | 1.42 (0.75-2.67) | 0.278 |
| AJCC7th staging | | | | | | | | |
| I | Ref | | Ref | | Ref | | Ref | |
| II | 6.62 (1.43–30.66) | 0.016 | 12.82 (2.50–65.70) | 0.002 | 7.44 (1.63–33.94) | 0.010 | 15.74 (3.17-78.08) | 0.001 |
| III | 23.57 (5.70–97.50) | <0.001 | 40.38 (8.27–197.15) | <0.001 | 33.72 (8.21–138.44) | <0.001 | 55.07 (11.52-263.31) | <0.001 |
| Adjuvant chemotherapy | | | | | | | | |
| No | Ref | | Ref | | Ref | | Ref | |
| Yes | 2.11 (1.16–3.84) | 0.015 | 0.40 (0.20–0.80) | 0.009 | 2.50 (1.44–4.34) | 0.001 | 0.27 (0.14-0.51) | <0.001 |

*HR* hazard ratio, *CI* confidence interval, *AJCC* American Joint Committee on Cancer, *BMI* body mass index (calculated as weight in kilograms divided by height in meters squared), *ECOG PS* Eastern Cooperative Oncology performance status, *ICG* indocyanine green.

significantly better than that of pN+ patients in the non-ICG group (log-rank *P* = 0.012; 3-year DFS rate: 68.0% vs 48.6%).

Among the full cohort, there are no significant interactive effects between ICG and LN dissection compliance on OS and DFS (*P*-interaction for OS = 0.077, adjusted *P*-interaction for OS = 0.061; *P*-interaction for DFS = 0.125, adjusted *P*-interaction for DFS = 0.094). While among the patients with pN+ stage disease, there was a significant interactive effect of ICG and LN dissection noncompliance on OS and DFS (*P*-interaction for OS = 0.033, adjusted *P*-interaction for OS = 0.028; *P*-interaction for DFS = 0.039, adjusted *P*-interaction for DFS = 0.033) (Supplementary Table 6).

## Discussion

This RCT aimed to evaluate the role of ICG in LN tracing during laparoscopic radical gastrectomy. Our study shows that laparoscopic

ICG fluorescence imaging-guided lymphadenectomy can improve the long-term OS and DFS of patients with GC and reduce the cumulative recurrence rate compared with conventional lymphadenectomy. This finding provides further evidence of the effectiveness and importance of ICG fluorescence imaging-guided lymphadenectomy in the treatment of GC.

Previous studies have shown that within the specified scope of dissection, the greater the number of LNs retrieved, the better the long-term survival of GC patients[11–14]. Therefore, for patients with GC, especially those with locally advanced GC, complete dissection of metastatic LNs and reduction of missed dissection of metastatic LNs are of great significance for accurate staging and subsequent treatment options. However, owing to the complex anatomy of the perigastric vascular fascia and the lack of tactile feedback in laparoscopic surgery, it is subjective to make decisions and evaluations based only

on the experience of the surgeon. Performing an efficient and accurate lymphadenectomy without increasing intraoperative complications is still a great challenge for surgeons, especially junior-trained ones. Because near-infrared ICG fluorescence can display the contour and boundary of perigastric LNs in real-time, it is helpful to guide surgeons to perform lymphadenectomy in the proper dissection plane more accurately. Several studies and systematic reviews, including this prospective study[5,10,15,16], have confirmed that ICG fluorescence imaging technology can guide surgeons to efficiently harvest more LNs in laparoscopic radical gastrectomy, which has attracted considerable attention from surgeons. However, the most striking question is whether ICG-guided lymphadenectomy is indeed associated with improved survival, and this still lacks long-term survival data[17].

This clinical trial found that the 3-year DFS rate and 3-year OS rates in the ICG group were significantly better than those in the non-ICG group, which may be due to more extensive and complete LN dissection in the ICG group. The pN stage of resectable GC is directly related to the number of metastatic LNs[18,19]. Moreover, we should not ignore the fact that there were two schools of thought in GC surgery with regards to LNs - one is that retrieving more LNs results in better staging and thus better clinical decisions about whether adjuvant chemotherapy is necessary[11]. The other is that routine H&E examinations cannot accurately evaluate LN micrometastasis, which is closely related to the poor prognosis of patients[20–22]. In other words, the more LNs are removed, the more positive and negative LNs with possible micrometastasis will increase. Dissecting a sufficient number of LNs in the standard lymphadenectomy area is necessary for accurate disease staging and avoiding missed dissection of metastatic LNs[23], thus having a positive impact on the prognosis of patients. Our data revealed that patients with total retrieved LNs ≥30 had a better prognosis than patients with total retrieved LNs <30, while all patients in the ICG group had a retrieved LN count of ≥30. Further stratified analysis showed that the prognosis of patients in the ICG group was better than that for patients in the non-ICG group for patients with total retrieved LNs ≥30.

Previous studies have shown that LN dissection noncompliance, especially major LN dissection noncompliance, significantly affects the long-term survival of GC patients[24–26]. With ICG imaging guidance, the surgeon can also evaluate the completeness of lymphadenectomy by imaging the residual LNs within the scope of dissection to effectively reduce LN dissection noncompliance. Our results also showed that among patients with LN metastasis, the 3-year DFS of patients who underwent compliant lymphadenectomy was significantly better than that of patients who underwent noncompliant lymphadenectomy. In contrast, pN+ patients in the ICG group had a better prognosis than pN+ patients in the non-ICG group, and there was a significant interactive effect of ICG and LN dissection noncompliance on long-term survival. It is suggested that ICG-guided LN dissection may achieve survival benefits by increasing the number of retrieved LN as well as reducing the LN dissection noncompliance rate of GC patients with LN metastasis.

It should be noted that the fluorescent LNs can only indicate, with an accuracy of about 62.2%–97.2%, that the LN receives lymphatic reflux from the tumor, though it is not necessarily a metastatic LN. Nevertheless, it is possible to have false negatives in ICG fluorescence, that is, nonfluorescent LNs with metastatic LNs as observed by near-infrared (NIR) imaging, with an incidence of 46.4% to 60%[6,8,27,28]. A possible reason for the false negative result is large-scale cancer invasion of LNs or lymphatics blocked by cancer cells. In this case, the tracer we used could not accumulate in the metastatic LNs[29,30]. Therefore, ICG fluorescence technology is mainly used to assist LN dissection, but cannot be used to determine LN metastasis. Furthermore, during surgery for GC, particularly in locally advanced GC, improper manipulation of lymphatic adipose tissue often leads to the release of free cancer cells from the lymphovascular pedicle and metstatic LN, thereby increasing the risk of recurrence[31,32]. Given that

ICG can track lymphatics and LNs well under high-resolution laparoscopic imaging (Supplementary Fig. 9), it may reduce the dissemination of free cancer cells caused by improper operations, such as incorrect handling of LNs containing tissue by surgeons to a certain extent. This technology can better reflect the tumor-free principles of surgical oncology.

The recent therapeutic effectiveness of robot-assisted gastrectomy guided by ICG has been reported[5,33–35]. While the application of ICG in open gastrectomy is focused on early-stage GC sentinel LNs research[27,36]. Moreover, the short-term and long-term efficacy of laparoscopic surgery has been shown to be non-inferior to that of open surgery[37–39]. However, whether the oncological effectiveness of robot-assisted gastrectomy is non-inferior to that of laparoscopic gastrectomy remains unclear and should be analyzed in future large-sample randomized controlled trials. Therefore, this study enrolled patients with GC who underwent laparoscopic surgery.

ICG solution was injected into the submucosal layer of the four quadrants around the primary tumor via endoscopy 1 day preoperatively. Patients who have previously undergone gastrectomy (such as distal gastrectomy) or endoscopic submucosal dissection may experience an alteration in their gastric wall anatomy, physiological function, and lymphatic drainage[7,40,41], which could change the lymphatic drainage pathway and affect the visualization effect of ICG to some extent. To ensure the homogeneity of the study population and not increase potential confounding variables, patients with a history of previous gastrectomy, endoscopic mucosal resection, or endoscopic submucosal dissection were excluded from this study.

With the reporting of previous studies[42–45], numerous guidelines, including the *Japanese Gastric Cancer Treatment Guidelines*, recommend neoadjuvant chemotherapy combined with surgical radical resection, rather than surgery only for patients with GC with bulky LNs[45,46]. Evidently, neoadjuvant chemotherapy affects the prognosis and subsequent treatment decisions of patients with GC[47,48], leading to significant heterogeneity in the study population. Therefore, we excluded patients with enlarged or bulky regional LNs with a diameter of >3 cm in our study. Additionally, we will conduct ICG-related research on patients with LN enlargement to explore the role of ICG fluorescence imaging technology in lymphadenectomy for such patients. For instance, our center is currently conducting an RCT study of ICG for patients with GC receiving neoadjuvant treatment (NCT04611997).

The previously reported safety results[10] indicated that the incidences of postoperative complications and postoperative recovery were comparable between the ICG and non-ICG groups. In addition, ICG can guide surgeons to harvest more LNs (50.5 vs. 42.0) and effectively reduce LN dissection noncompliance (31.8% vs. 57.4%) in laparoscopic radical gastrectomy for GC. Taken together, the short- and long-term results of this study showed that ICG tracer-guided laparoscopic radical gastrectomy performed by expert surgeons in China's high-volume referral center is superior to conventional naked laparoscopic gastrectomy, especially for patients with locally advanced GC. However, the generalizability of these findings to practice settings where staging, surgical training, and use of adjuvant therapy are different may be limited.

The present study had several limitations. First, this study only included patients from a single center. Based on the findings of this RCT, the Chinese Laparoscopic Gastrointestinal Surgery Study (CLASS) group conducted a multicenter RCT (CLASS-11 trial; NCT04593615) to provide further evidence. Second, the study did not include patients receiving neoadjuvant therapy, and patients often had tumor and LN regression and fibrotic responses after neoadjuvant therapy, although previous studies have shown that ICG tracing can also improve the number of LNs retrieved in patients who received neoadjuvant chemotherapy[49]. Third, the study was conducted in China, so it is not clear whether the results could be generalized to

Western settings[50]. In addition, this study adopted a post-hoc analysis based on RCT to attempt to explain the survival reasons for ICG patients' benefits. Therefore, caution should be exercised when promoting the conclusions of this study. Moreover, this study was performed at an institution with rich experience in GC surgery. At last, we look forward to future research on intraoperative LN metastasis prediction using techniques, such as fluorescence intensity and specific antibody-labeled fluorescent dyes. However, ICG-guided laparoscopic radical gastrectomy may provide greater assistance to junior-trained gastric surgeons.

In conclusion, for patients with resectable GC, ICG fluorescence imaging-guided lymphadenectomy can not only significantly improve the total number of LNs retrieved in laparoscopic D2 radical gastrectomy for GC, but it also shows substantial long-term oncological efficacy compared with conventional lymphadenectomy. We suggest that ICG-guided laparoscopic radical lymphadenectomy for GC be routinely performed.

## Methods

### Study design

The current study was a phase 3, parallel, open-label RCT conducted at the Fujian Medical University Union Hospital (FMUUH), a tertiary referral teaching hospital in China. This clinical trial was registered at https://www.clinicaltrials.gov before patient enrollment (clinical trial identifier NCT03050879). This study was approved by the institutional review board of FMUUH (IRB number: 2016YF015-02) and was conducted in accordance with the Declaration of Helsinki. The original study protocol is available in the Supplementary Information in Supplementary Note 2.

### Participants

Patients were eligible to participate if they were aged 18 to 75 years, had an Eastern Cooperative Oncology Group (ECOG) score of 0 (asymptomatic) or 1 (symptomatic but completely ambulatory), and had histologically confirmed gastric adenocarcinoma diagnosed at the preoperative clinical stage of cT1 to cT4a, N0/+, M0 according to the 7th Edition of the American Joint Committee on Cancer (AJCC) Staging Manual[51]. Patients were excluded if they had enlarged or bulky regional LNs with a diameter of more than 3 cm as measured by preoperative imaging, had a history of allergy to iodine agents, or had a history of previous gastrectomy, endoscopic mucosal resection, or endoscopic submucosal dissection. The detailed eligibility criteria are shown in Supplementary Table 7. The first patient was enrolled on November 19, 2018, and the last was recruited on July 13, 2019.

### Randomization and masking

Eligible patients were randomly assigned by a 1:1 ratio to either the ICG or non-ICG group. The data manager (M.L.), who was not involved in the eligibility assessment and recruitment of patients, performed randomization with a list of randomly ordered treatment identifiers generated by a permuted block design using SAS (version 9.2; SAS Institute Inc.). The allocation sequence was concealed from the surgeons who enrolled the patients until they were formally randomized to their groups. Informed consent was given to eligible patients two days before the operation. Either patients assigned to ICG or non-ICG groups, preoperative endoscopy is necessary for tumor location one day before the operation. The difference is that the ICG group received drug injections but the non-ICG group did not. Although it was not feasible to blind the surgeons and participants owing to the nature of the surgical clinical trial, the chemotherapy-treating oncologists were unaware of the intervention received by the patients.

### Procedures

In the ICG group, ICG was endoscopically injected around the tumor in patients one day before operation; 1.25 mg/mL ICG (Dandong

Yichuang Pharmaceutical Co) was prepared in sterile water, and 0.5 mL of the solution was injected into the submucosal layer at four quadrants around the primary tumor, amounting to 2.5 mg of ICG. We used the PINPOINT Endoscopic Fluorescence Imaging System (NOVADAQ, Stryker, US) equipped with a fluorescence mode to obtain NIR fluorescent images in the ICG group. A simple finger-click can convert visible light into NIR images (infrared imaging, green fluorescence, and color-segmented fluorescence) without the need to change any equipment. Intraoperatively, the fluorescent mode was switched according to the situation (Supplementary Fig. 10). In the ICG group, during the surgical procedure, the surgeons tended to utilize the green fluorescence imaging mode to perform LN dissection. If necessary, they switched between white light and green fluorescence imaging modes to observe the surgical area. Following the LN dissection in each area, we also employed fluorescence imaging to assess the completeness of the LN dissection.

All the operations were performed by two surgeons (C.-H.Z. and C.-M.H.) who are members of the same surgical team. All the participating surgeons in our study met the following criteria: they had performed more than 100 laparoscopic radical gastrectomies, completed a learning curve in laparoscopic radical LN dissection, passed the blind surgical video examination, and had ample experience in ICG-guided LN dissection for GC. The surgeons were unaware of the specific allocation of the patient before the start of surgery, to prevent any potential discrimination of surgical strategy.

All pathological evaluations were performed in a standard manner. For the pathological evaluation protocol, we referred to the *GASTRIC CANCER STRUCTURED REPORTING PROTOCOL (2nd Edition, 2020)*[52]. After resecting the specimens, the surgeons positioned each LN station according to the location of the blood vessel clips retained in the specimens during the operation and sorted each LN station according to the *Japanese Research Society for Gastric Carcinoma criteria*[53]. Surgeons examined all the specimens. The specimens were immediately sent to the department of pathology after repacking, and the LNs of each station were examined by two or more experienced pathologists by palpation and microscopy.

LNs containing isolated tumor cells, defined as single tumor cells or small clusters of cells ≤0.2 mm in greatest diameter, without stromal reaction, are classified as pN0 in GC[54]. There is no micro-metastasis (N1mi) category in staging GC[54]. LNs containing clusters of cells >0.2 mm in diameter are considered positive. In pretreated GCs, positive LNs are defined as having at least one focus of residual tumor cells in the LNs regardless of size. LNs with acellular mucin pool or fibrotic LNs with no viable tumor are considered negative. All LNs were bisected and evaluated without routine serial sectioning.

Information regarding hematoxylin-eosin staining of paraffin sections includes the following steps:

(1) Tissue embedding. (1) ethanol dehydration: tissues are dehydrated gradually with ethanol solutions of different concentrations (70%, 80%, 90%, 95%, 100%, and 100%) for 40 min each. (2) clearing: the tissues are immersed in three xylene baths, 1 h for each bath. (3) impregnation: the tissues are immersed in three paraffin baths, 1 h for each bath. (4) embedding: liquid paraffin is poured into a mold box, and the tissue block that has been impregnated with paraffin is laid flat on the bottom. Notably, the cutting surface should be placed facing downwards. After the paraffin has solidified, the embedding frame is removed. Once the tissue block has cooled down and becomes completely hard, the excess paraffin around the tissue is trimmed and kept moderately to facilitate sectioning.

(2) Section preparation: the pre-cooled wax block is fixed onto the microtome, ensuring that the section of the wax block is parallel to the blade, which is typically tilted at 15°. The wheel advance mechanism is rotated and slice thickness is adjusted to 4 μm to obtain evenly thick slices. A brush is held in the left hand and the microtome handle is rotated with the right hand to cut the slices. The slices are gently lifted

with the brush and the excess wax is tweezed. The slices are placed face up in the water bath of the slide warmer, which is set at a temperature of approximately 45 °C. After flattening the slices, they are picked with forceps. The slices are attached to the glass slides by immersing one end of a slide vertically in the water and the forceps are used to push the slice two-thirds of the way onto the slide. After attaching the slices to the slides, they are left to air dry and then placed in a slide warmer at 65 °C for 1 h, followed by a 2-h baking process in an oven.

(3) Deparaffinization of paraffin-embedded tissue sections: the paraffin sections are embedded in xylene I for 10 min, followed by xylene II for 10 min, and xylene III for 10 min. Then, the sections are gradually deparaffinized in a series of solutions: anhydrous ethanol I for 5 min, anhydrous ethanol II for 5 min, 90% ethanol for 5 min, 80% ethanol for 5 min, 70% ethanol for 5 min, and finally 50% ethanol for 5 min.

(4) HE staining: the paraffin sections were stained with hematoxylin for 0.5–1 min, rinsed with tap water, differentiated in 1% hydrochloric acid alcohol for a few seconds, rinsed with tap water, blue with 1% ammonia water for 1 min, rinsed with running water for a few seconds, and counterstained with eosin for several seconds, followed by rinsing with running water.

(5) Dehydration and mounting of slides: the paraffin sections were sequentially immersed in 75% ethanol for 2 min, 85% ethanol for 2 min, and then in absolute ethanol for 5 min twice. Subsequently, the sections were cleared in xylene for 5 min and mounted with neutral gum after being removed from the xylene bath.

(6) Interpretation of results: the cell nucleus is blue, and the cytoplasm is red.

The extent of gastric resection and D2 lymphadenectomy was determined according to the tumor location, as indicated in the Japanese guidelines[55]. After lymphadenectomy in the ICG group, NIR imaging was routinely performed for the final observation of residual fluorescent LNs, and any remaining stained nodes were removed. Adjuvant chemotherapy (6 months of a fluorouracil-based chemotherapy regimen) was recommended for patients with pathologic stage II or greater advanced disease[51], with the choice of a specific regimen and treatment duration at the discretion of the treating oncologist.

## Outcomes

The primary original protocol endpoint (Supplementary Information 2) was the total number of retrieved LNs[10], and the secondary endpoints of the original study and prior report and for this current report the primary endpoints were the 3-year OS, 3-year DFS, and recurrence patterns. Outcomes pertaining to safety and efficacy, including diagnostic sensitivity and specificity, postoperative recovery course, morbidity, and mortality rates, have been previously reported[10].

The current AJCC Staging manual recommends that the removal of ≥30 regional LNs is desirable[54]. A reference number of 30 was used. LN dissection noncompliance was defined as the absence of LNs from more than 1 LN station that should have been excised[24,25].

OS was defined as the time from surgery to death from any cause or the last follow-up, and DFS was defined as the time from surgery to recurrence or death from any cause or the last follow-up.

A minimum follow-up period of 36 months was required for each patient after operation. Follow-up was conducted every 3 months for the first 2 years postoperatively, and every 6 months for the next 3 years.

Most routine follow-up appointments included (1) physical examination and blood testing with carcinoembryonic antigen, cancer antigen 12-5, and cancer antigen 19-9 every 3 months for the first 2 years and every 6 months thereafter; (2) chest X-ray and abdominal computed tomographic scans every 6 months for 3 years; and (3) annual upper gastrointestinal endoscopy for 3 years. 18F-fluorodeoxyglucose positron emission tomography/computed tomography (18F-FDG PET/CT) was recommended if recurrence was suspected. Recurrence was identified based on medical history and physical examination in combination with imaging evaluation, cytology, or tissue biopsy (preferred when feasible). Otherwise, patients attended follow-up visits at shorter intervals than the planned schedule. Patients with specific symptoms, such as abdominal mass, weight loss, or obstruction that could develop concurrently with recurrence were evaluated, regardless of their follow-up schedule.

## Sample size

The main evaluation index in this study was the total number of retrieved LNs. Based on previous studies[56–58], the total number of retrieved LNs was 32.9 in the control group. A sample size of 107 patients per group was calculated as necessary for an α of 0.05, power of 80%, and margin delta of 15%. Assuming a dropout rate of 20%, at least 133 cases were required for each group. The nQuery Advisor 7.0 (Cork, Ireland) was used to calculate the sample size.

## Statistical analysis

There are no deviations in the analysis plan compared with the pre-registered protocol. The per-protocol analysis set population was used for all analyses. ITT analysis was conducted for the primary end point and secondary survival points only. Continuous variables are expressed as mean (standard deviation (SD)), and categorical variables are expressed as numbers. The differences between the groups were assessed using the $t$-test or $\chi^2$ test, as appropriate. All tests were two-sided, with a significance level set at $P < 0.05$.

The 3-year DFS and OS rates were calculated using the Kaplan-Meier method, and the log-rank test was used to determine significance. The hazard ratios (HRs) comparing the ICG and non-ICG groups were estimated using Cox regression after confirmation of the proportional hazards assumption. Multivariable Cox regression analyses were performed to evaluate the effect of operation type on survival, after adjustment for clinicopathologic covariables that were significantly associated with the outcome in univariable analyses. Factors with a $P$-value < 0.05 in the univariable analysis will be included in further multivariable analysis.

All-cause death was treated as a competing event for recurrence. The cumulative incidence in the presence of competing risks was calculated, and competing-risk survival regression was used as an alternative to Cox regression[37,59–61]. Multivariable Cox regression was used for recurrence and all-cause death, after adjustment for sex, AJCC7th staging, and adjuvant chemotherapy. $P$ for multiplicative interactions were investigated[62,63].

All data were analyzed using SPSS statistical software, version 22.0 (SPSS Inc), and the R software environment, version 4.2.0 (R Foundation for Statistical Computing). Statistical analysis was performed from July to October 2022. Supplementary Data 1 contains the clinical data of this study.

## Reporting summary

Further information on research design is available in the Nature Portfolio Reporting Summary linked to this article.

# Data availability

The data supporting the findings in this study are available under controlled access due to data privacy laws related to patient consent for data. All the original clinical data will be made available on request from the corresponding author (Huang CM) at any time in a de-identified manner for research purposes only. The remaining data are available within the Article, Supplementary Information. Requests for data sharing will be managed in accordance with Fujian Medical University Union Hospital's data access and sharing policy, which can be

found in Supplementary Note 1. However, it must be ensured that necessary agreements are enforced, such as those regarding security, patient privacy, and the consent for specified data use, in line with the constantly evolving applicable data protection laws. The original study protocol is available in the Supplementary Information in Supplementary Note 2. Source data are provided with this paper.

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

## Acknowledgements

We thank those who have devoted a lot to this study, including pathologists, further-study doctors, statisticians and nurses. Thanks for Dr. Zhi-Hong Huang, Public Technology Service Center, Fujian Medical University. Thank you to statisticians Liu Fengqiong and Chen Fa for their guidance in statistics. This study was supported by the Fujian Province Medical "Creating high-level hospitals, high-level medical centers, and key specialty projects (Min [2021] No.76), Talent Initiation Fund Project of Fujian Medical University Union Hospital (2022XH041), Excellent Young Scholars Cultivation Project of Fujian Medical University Union Hospital (2022XH021). The funding source had no role in the design and conduct of the study; collection, management, analysis, and interpretation of the data; preparation, review, or approval of the manuscript; and decision to submit the manuscript for publication.

## Author contributions

C.Q.Y., Z.Q., H.C.M. and Z.C.H.: Concept and design. L.Z.Y., L.P., X.J.W., J.M.C., W.H.G. and L.G.T.: Acquisition, analysis, or interpretation of data. C.Q.Y., Z.Q., L.Z.Y., H.C.M. and Z.C.H.: Drafting of the manuscript. C.Q.Y., Z.Q., L.Z.Y., H.C.M. and Z.C.H.: Statistical analysis. Z.Q.L., W.J.B., L.J.X., L.J., C.L.L., L.M., T.R.H., H.Z.N., Z.G.R., H.X.B., W.H.G., L.Y.F. and X.K.X.: Administrative, technical, or material support. Z.Q. and C.Q.Y.: Supervision.

## Competing interests

The authors declare no competing interests.

## Additional information

Qi-Yue Chen[1,2,6], Qing Zhong [1,2,3,6], Zhi-Yu Liu[1,2,3,6], Ping Li[1,2,3], Guang-Tan Lin[1,2,3], Qiao-Ling Zheng[4], Jia-Bin Wang[1,2,3], Jian-Xian Lin[1,2,3], Jun Lu[1,2,3], Long-Long Cao[1,2,3], Mi Lin[1,2,3], Ru-Hong Tu[1,2,3], Ze-Ning Huang[1,2,3], Gui-Rong Zeng[5], Mei-Chen Jiang[4], Hua-Gen Wang[1,2,3], Xiao-Bo Huang[1,2,3], Kai-Xiang Xu[1,2,3], Yi-Fan Li[1,2,3], Chao-Hui Zheng [1,2,3] ✉, Jian-Wei Xie[1,2,3] ✉ & Chang-Ming Huang [1,2,3] ✉

[1]Department of Gastric Surgery, Fujian Medical University Union Hospital, Fuzhou, China. [2]Department of General Surgery, Fujian Medical University Union Hospital, Fuzhou, China. [3]Key Laboratory of Ministry of Education of Gastrointestinal Cancer, Fujian Medical University, Fuzhou, China. [4]Department of Pathology, Fujian Medical University Union Hospital, Fuzhou, China. [5]Diagnostic Pathology Center, Fujian Medical University, Fuzhou, China. [6]These authors contributed equally: Qi-Yue Chen, Qing Zhong, Zhi-Yu Liu. ✉e-mail: wwkzch@163.com; xjwhw2019@163.com; hcmlr2002@163.com

