## [Peer Review File · Nature Communications]

Reviewers' Comments:

Reviewer #2:

Remarks to the Author:

A prospective study was designed in which subjects with gastric cancer (T1-4a, N0/+, M0) were allocated to the control or fluorescence (FLU)-imaging groups at a 1:1. The authors assessed the long-term (3-years) oncological outcome (OS, DFS, recurrence) (secondary objectives) of ICG-FLU based lymph node harvesting in subjects undergoing laparoscopic gastrectomy in addition to total LN yield (primary objective).

Introduction

1. Line 113 - what is referred to with visual instruments? Blue dyes historically have been used for LN mapping. Also, starting line 120 the authors mention that ICG has been adapted, but that outcome studies are lacking and therefore that this study was performed. Please consider rewording this sentence omitting "without the aid of visual instruments".
2. Line 118 - what is meant with minimally invasive surgical equipment? Do the authors mean laparoscopic cameras? Or...
3. Please explain "LN noncompliance" (line 130).

Methodology

4. Line 149 - why were subjects with large LNs excluded from the study? From sentinel node studies in other indications we know that when large metastasis are present in LNs, that this sometimes results in lymphatic drainage being rerouted to a neo-sentinel node simply because the tracer (ICG for example) cannot reach. Although I would agree that it's hard to miss a 3 cm LN during surgery, if any lymphatic drainage is rerouted because of that could this not possibly result in finding additional LNs that would normally not have been detected (which thus possibly further improves patient outcome and decreases the risk for recurrence)? See also the comments of the authors in the discussion (line 399 and further) -- would this not "urge" to also investigate the approach in those patients with known large mets to demonstrate its value?
5. Line 151 - why were subjects with a history of previous gastrectomy, endoscopic mucosal resection, or endoscopic submucosal dissection excluded from the study? Is there evidence that ICG-based LN mapping does not work in patients that underwent prior surgical procedure? For sentinel node studies in e.g. penile cancer it has been demonstrated that even in previously treated patients, repeat sentinel node biopsy is possible and very valuable [studies performed by a.o. Simon Horenblas at the Dutch Cancer Institute].
6. Line 153 - do you think there could be a possible bias to surgeons knowing that their subject did (not) get the ICG injection and therefore that they would prepare the case more precisely by additional study of the preoperative imaging data as such to get a better sense of the possible location of LNs?
7. Line 170 - please specify the camera that you used for this study, was it the Pinpoint?
8. Line 170 - which imaging mode was preferred by the surgeon? The green fluorescence? The black and white? or the color-segmented mode? Why?
9. Line 176 - which team? How many surgeons were involved?
10. Line 193 - how does noncompliance occur? Is this simply because insufficient fatty tissue possibly harbouring nodes is removed?
11. Line 205 - 18F-FDG pet? please specify
12. What was the intraoperative imaging protocol? Was FLU imaging performed on the fly? Or was it performed after clearing out each respective station to assess for any remaining LNs etc.?

Results

13. Line 241 - what is meant with ICG contamination? How was / is contamination determined?
14. Line 248 - do the authors think that the tumor location is / was of effect on the LN yield in each group?
15. Line 249 and line 258 and line 272 - despite the LN yield being much higher in the ICG group, adjuvant chemo rates were similar between the groups. Though the OS and DFS is better in the ICG group. Does this mean that adjuvant chemo has no effect on the treated patients, but the number of nodes harvested does? Could all be explained by the fact that in a fair number of subjects (12%) in the control arm fewer than 30 nodes were harvested (line 295)?
16. Should like 302 be "less than 30 retrieved LNs?"

17. 30+% of non-compliance generally speaking seems really high to me. Can the authors elaborate on how noncompliance occurs? Is it because its hard to remove tissue here? Or is it too risky to remove tissue from certain stations? Or is it that if visually no nodes are seen nothing will be removed?

18. There seems to be no significant differences in OS between compliant and non-compliant subjects, only in DFS, but only in the pN+ patients, although recurrence rates in these subjects were not significantly different between the compliant and non-compliant subjects. Though in the N0 groups OS and DFS rates are significantly different. How do the authors explain this?

Discussion:

19. Line 376 - this is new and comes out of the blue, particularly also because nothing about the H&E staining protocol is described. Please include the pathology assessment protocols into the methods section. Are nodes bisected? Are nodes serially sectioned and then evaluated? Etc. The more levels are collected for any given node the higher the likelihood of finding (micro-) mets, however it comes at the cost of time.

20. In following my comment of line 376 - have the authors considered triaging ICG+ LNs based on their fluorescence intensity to be able to predict those nodes that are at risk of harboring mets? Albeit slightly different Nishio et al., in Nature Comm 2019 described such an approach for an antibody-IRDye800.

21. Line 408-ish - what about in-transit LN mets?

Figures/Tables

22. Did the authors look at the relationship between BMI and noncompliance in the ICG-group?

General questions:

23. ICG-based LN harvesting is a rather universal technology and has successfully been implemented during both open, and (robot-assisted laparoscopic procedures. Why were only subjects that were operated laparoscopically included?

24. How many surgeons were involved in this study? Were the surgeons all experienced with gastrectomy and LN mapping. Were they all experienced in the use of ICG for LN mapping? If not, did you find any differences in LN yield between naive and experienced surgeons with either technology?

Reviewer #3:

Remarks to the Author:

This article reports the long-term outcomes from a study in gastric cancer patients randomized to receive either ICG fluorescence imaging vs non-ICG. The primary endpoint of the original study was the number of lymph nodes retrieved and this analysis has been already published. The current article now presents the secondary objectives of the original study: survival outcomes and recurrence patterns for these patients. The paper should state earlier and clearer that the reported analyses are based on the original randomized study.

There are a lot of subset analyses reported in this paper that do not appear to have been prespecified in the original protocol. This not only increases the number of tests performed and thus increases the false discovery rate, but also could be questioned as data driven.

It was not explained why the ITT analysis was not attempted: was peritoneal metastasis part of the exclusion criteria? Why were these patients excluded? Also, for the survival analyses, withdrawn from study patients could have been censored.

It is not specified for the multivariable analysis, what screening threshold was used in the univariable regressions to be included in the multivariable analyses.

More details should be provided about the method (and reference for it) used for the competing risk analysis. There were 2 p-values included in Table 2 and they have not been properly explained.

There were some comparisons made for the overall study population (combining the 2 arms:

eFigures 4 and 6). This has not been justified. Furthermore, an interaction between ICG and LN compliance has been assessed among patients with pN+, however it would have been helpful to evaluate such interactions for other variables for which instead stratified analyses have been performed; for example, interaction between ICG arm and number of retrieved lymph nodes (continuous), between ICG arm and LN compliance in the full population, etc.

Discussion claims that "...for the same pT stage, the possibility of detecting LM metastasis in the ICG group was higher than that in the non-ICG group" (eFigure 3) however no p-value was provided and no method referenced.

Some comparisons were highlighted when they had a significant pvalue, but some non-significant pvalues were not mentioned (for example eTable 3, OS between ICG vs non ICG among patients with 30 or more lymph nodes retrieved. There were only 16 patients that had less than 30 nodes retrieved so when comparing ICG vs non-ICG in the full population (significant pvalue) vs only among those with ≥ 30 nodes (pvalue is no longer statistically significant) it appears that these 16 patients with less than 30 nodes play a role.

Absolute differences and risk differences are reported however risk difference was defined only in the legend of a table. I would recommend removing those quantities or providing more details and reference.

The hazard ratios for the competing risk analysis were above 1 and that contradicts the interpretations in the paper. Has the reference group been switched?

Cumulative incidence of locoregional recurrence is listed but no timepoint provided in the paper at which they were evaluated.

The log-rank test compares the overall curves not a specific timepoint as implied in some parts (for example in Abstract).

Reviewer #4:
None

RESPONSE TO REVIEWERS

Title: **“Long-Term Outcomes of Indocyanine Green Fluorescence Imaging-Guided versus Conventional Laparoscopic Lymphadenectomy for Gastric Cancer”** We are extremely grateful to the editor and anonymous reviewer for their valuable comments and suggestions, which have helped improve the quality of the manuscript. We have studied the reviewers' comments and have made corresponding modifications and corrections, which we hope will meet their approval. We have revised the manuscript according to the referee's suggestions. Our descriptions of the revisions are as follows.

REVIEWER COMMENTS

Reviewer #2 - Image guided surgery - (Remarks to the Author):

A prospective study was designed in which subjects with gastric cancer (T1-4a, N0/+, M0) were allocated to the control or fluorescence (FLU)-imaging groups at a 1:1. The authors assessed the long-term (3-years) oncological outcome (OS, DFS, recurrence) (secondary objectives) of ICG-FLU based lymph node harvesting in subjects undergoing laparoscopic gastrectomy in addition to total LN yield (primary objective).

Introduction

1. Line 113 - what is referred to with visual instruments? Blue dyes historically have been used for LN mapping. Also, starting line 120 the authors mention that ICG has been adapted, but that outcome studies are lacking and therefor that this study was performed. Please consider rewording this sentence omitting "without the aid of visual instruments".

The authors' answer: Thank you for your insightful feedback. We apologize for any confusion caused. As per your request, we have removed the sentence

“without the aid of visual instruments”.

2. Line 118 - what is meant with minimally invasive surgical equipment?

Do the authors mean laparoscopic cameras? Or...

The authors' answer: Thank you for your valuable comment. We apologize for any confusion caused. We have modified to the terminology, changing 'minimally invasive surgical equipment' to 'minimally invasive surgery' for better comprehension by the readers.

3. Please explain "LN noncompliance" (line 130).

The authors' answer: Thank you for your valuable comment. We apologize for any confusion caused. According to the references¹⁻³, within the scope of D2 dissection, LN dissection noncompliance was the absence of LNs that should have been resected from more than one LN station. We have added the contents to the *Method* of the revised manuscript.

Reference

1. De Steur WO, Hartgrink HH, Dikken JL, et al. Quality control of lymph node dissection in the Dutch Gastric Cancer Trial. *Br J Surg* 2015; 102(11):1388-93.
2. Chen QY, Xie JW, Zhong Q, et al. Safety and Efficacy of Indocyanine Green Tracer-Guided Lymph Node Dissection During Laparoscopic Radical Gastrectomy in Patients With Gastric Cancer: A Randomized Clinical Trial. *JAMA Surg* 2020; 155(4):300-311.
3. Chen QY, Zhong Q, Liu ZY, et al. Does Noncompliance in Lymph Node Dissection Affect Oncological Efficacy in Gastric Cancer Patients Undergoing Radical Gastrectomy? *Ann Surg Oncol*. 2019 Jun;26(6):1759-1771.

Methodology

4. Line 149 - why were subjects with large LNs excluded from the study?

From sentinel node studies in other indications we know that when large metastasis are present in LNs, that this sometimes results in lymphatic

drainage being rerouted to a neo-sentinel node simply because the tracer (ICG for example) cannot reach. Although I would agree that its hard to miss a 3 cm LN during surgery, if any lymphatic drainage is rerouted because of that could this not possibly result in finding additional LNs that would normally not have been detected (which thus possible further improves patient outcome and decreases the risk for recurrence)? See also the comments of the authors in the discussion (line 399 and further) -- would this not "urge" to also investigate the approach in those patients with known large mets to demonstrate its value?

The authors' answer: Thank you for your valuable comment. With the reporting of various large-scale prospective clinical studies¹⁻⁴, numerous guidelines, including the *Japanese Gastric Cancer Treatment Guidelines*, recommend neoadjuvant chemotherapy combined with surgical radical resection, rather than surgery only, for patients with gastric cancer (GC) with bulky LNs^{1,5}. Evidently, neoadjuvant chemotherapy affects the prognosis and subsequent treatment decisions of patients with GC⁶⁻⁷, leading to significant heterogeneity in the study population. Therefore, we excluded patients with enlarged or bulky regional LNs with a diameter of >3 cm in our study. Additionally, we will conduct ICG-related research on patients with LN enlargement to explore the role of ICG fluorescence imaging technology in LN dissection for such patients. For instance, our center is currently conducting an RCT study of ICG for patients with GC receiving neoadjuvant treatment (NCT04611997), and we look forward to future reports on the application of ICG in the LN enlargement patients with GC. We have added these contents to the *Discussion* in the revised manuscript.

Reference

1. Japanese Gastric Cancer Association. Japanese Gastric Cancer Treatment Guidelines 2021 (6th edition). *Gastric Cancer*. 2023 Jan;26(1):1-25.
2. Iwasaki Y, Terashima M, Mizusawa J, et al. Gastrectomy with or without

- neoadjuvant S-1 plus cisplatin for type 4 or large type 3 gastric cancer (JCOG0501): an open-label, phase 3, randomized controlled trial. *Gastric Cancer*. 2021 Mar;24(2):492-502.
3. Kang YK, Yook JH, Park YK, et al PRODIGY: a phase III study of neoadjuvant docetaxel, oxaliplatin, and S-1 plus surgery and adjuvant S-1 versus surgery and adjuvant S-1 for resectable advanced gastric cancer. *J Clin Oncol*. 2021;39:2903-13.
 4. Zhang X, Liang H, Li Z, et al Perioperative or postoperative adjuvant oxaliplatin with S-1 versus adjuvant oxaliplatin with capecitabine in patients with locally advanced gastric or gastroesophageal junction adenocarcinoma undergoing D2 gastrectomy (RESOLVE): an open-label, superiority and non-inferiority, phase 3 randomised controlled trial. *Lancet Oncol*. 2021;22(8):1081–92.
 5. Ajani JA, et al. Gastric Cancer, Version 2.2022, NCCN Clinical Practice Guidelines in Oncology. *Journal of the National Comprehensive Cancer Network : JNCCN* 20, 167-192 (2022).
 6. Katayama H, Tsuburaya A, Mizusawa J, et al. An integrated analysis of two phase II trials (JCOG0001 and JCOG0405) of preoperative chemotherapy followed by D3 gastrectomy for gastric cancer with extensive lymph node metastasis. *Gastric Cancer*. 2019 Nov;22(6):1301-1307.
 7. Takahari D, Ito S, Mizusawa J, et al; Stomach Cancer Study Group of the Japan Clinical Oncology Group. Long-term outcomes of preoperative docetaxel with cisplatin plus S-1 therapy for gastric cancer with extensive nodal metastasis (JCOG1002). *Gastric Cancer*. 2020 Mar;23(2):293-299.

5. Line 151-why were subjects with a history of previous gastrectomy, endoscopic mucosal resection, or endoscopic submucosal dissection excluded from the study? Is there evidence that ICG-based LN mapping does not work in patients that underwent prior surgical procedure? For sentinel node studies in e.g. penile cancer it has been demonstrated that even in previously treated patients, repeat sentinel node biopsy is possible and very valuable [studies performed by a.o. Simon Horenblas at the Dutch Cancer Institute].

The authors' answer: Thank you for your valuable comment. In our study, ICG solution was injected into the submucosal layer of the four quadrants around the primary tumor via endoscopy 1 day preoperatively. Patients who have previously undergone gastrectomy (such as distal gastrectomy) or

endoscopic submucosal dissection may experience alteration in their gastric wall anatomy, physiological function, and lymphatic drainage¹⁻³, which could change the lymphatic drainage pathway and affect the visualization effect of ICG to some extent. To ensure the homogeneity of the study population and not increase potential confounding variables, patients with a history of previous gastrectomy, endoscopic mucosal resection, or endoscopic submucosal dissection were excluded from this study. Our center has also conducted an ICG-related RCT specifically for patients with remnant GC (NCT05618821), and we have added this information in the *Discussion* of the revised manuscript.

Reference

1. Honda S, Bando E, Makuuchi R, et al. Effects of initial disease status on lymph flow following gastrectomy in cases of carcinoma in the remnant stomach. *Gastric Cancer*. 2017 May;20(3):457-464.
2. Shimada H, Fukagawa T, Haga Y, et al. Does remnant gastric cancer really differ from primary gastric cancer? A systematic review of the literature by the Task Force of Japanese Gastric Cancer Association. *Gastric Cancer*. 2016 Apr;19(2):339-349.
3. Roh CK, Choi S, Seo WJ, et al. Indocyanine green fluorescence lymphography during gastrectomy after initial endoscopic submucosal dissection for early gastric cancer. *Br J Surg*. 2020 May;107(6):712-719.

6. Line 153 - do you think there could be a possible bias to surgeons knowing that their subject did (not) get the ICG injection and therefore that they would prepare the case more precisely by additional study of the preoperative imaging data as such to get a better sense of the possible location of LNs?

The authors' answer: Thank you for your valuable comment. The two surgeons involved in this study are members of the same surgical team. Prior to the surgery, the surgeons were not informed regarding the specific patient allocation and only became aware of the group assignments during the operation. Both surgeons possess extensive surgical experience and have

undergone training on the learning curve of laparoscopic surgery. Regardless of whether or not an ICG injection was performed, they carefully reviewed each patient's imaging data preoperatively to ensure safety and thoroughly LN dissection. They did not conduct additional preparation on certain special patients. Therefore, the surgeons were unaware of the specific randomization group of the patient before the start of surgery to prevent any potential discrimination of surgical strategy. We have added these contents to the *Method* of the revised manuscript.

7. Line 170 - please specify the camera that you used for this study, was it the Pinpoint?

The authors' answer: Thank you for your valuable comment. We used the PINPOINT Endoscopic Fluorescence Imaging System (NOVADAQ, Stryker, US) equipped with the fluorescence mode to obtain near-infrared fluorescent images in this study. We have added these contents to the *Method* of the revised manuscript.

8. Line 170 - which imaging mode was preferred by the surgeon? The green fluorescence? the black and white? or the color-segmented mode? Why?

The authors' answer: Thank you for your valuable comment. The surgeons preferred using the green fluorescence mode, aided by the Pinpoint system, which allows for the synchronous overlaying of green fluorescent images to integrate white light and near-infrared images. This high-resolution mode effectively displays visible LNs without compromising surrounding tissues. After dissection of LNs, the surgeons can switch between white light and green fluorescence imaging modes to observe the operative area.

9. Line 176 - which team? How many surgeons were involved?

The authors' answer: Thank you for your valuable comment. The surgical team involved in this study was from the Department of Gastric Cancer at Fujian Medical University Union Hospital, consisting of two surgeons (C.-H.Z. and C.-M.H.). All participating surgeons in our study met the following criteria: they had performed more than 100 laparoscopic radical gastrectomies, completed a learning curve in laparoscopic radical LN dissection, passed the blind surgical video examination.

10. Line 193 - how does noncompliance occur? Is this simply because insufficient fatty tissue possibly harbouring nodes is removed?

The authors' answer: Thank you for your valuable comment. According to previous reports¹, within the scope of D2 dissection, LN dissection noncompliance was the absence of LNs that should have been resected from more than one LN station. Even with a standardized and systematic surgical procedure for GC LN dissection, missing microscopically involved LNs is possible. The reasons for LN dissection noncompliance are commonly related to several factors. First, in laparoscopic surgery, if the gastric tumor is large, it can compress the surgical field and affect the exposure of the stomach vessels and surrounding area of the lymphatic adipose tissue, which may result in missed LNs in this surgical area¹. Second, different surgical techniques and adhesions may hinder the complete removal of LNs². Third, numerous studies have reported that high BMI or increased intra-abdominal fat directly reduces the number of detected LNs³⁻⁷. Patients with high BMI often have a large amount of adipose tissue accumulated in the abdomen, making it difficult to distinguish the relationship among pancreatic tissue, adipose tissue, and LNs during surgery, which therefore makes LN dissection more difficult. Especially for patients with high BMI, specific areas, such as the celiac trunk and

pancreatic head, have very deep LNs, which further increases the difficulty of accurate positioning and dissection^{1,8}. Finally, the number of LNs considerably varies in each LN station in the normal gastric drainage LN area. Sometimes, some small stations (such as nos. 2, 5, and 10) may not contain any LNs at all, indicating biological variability⁹⁻¹⁰. Therefore, even if surgeons perform LN dissection for all LN stations according to the D2 criteria for radical resection of GC, there may be no LNs in the specimen. Considering this biological variability, the Dutch Gastric Cancer Trial study considered that LN dissection for GC allowed one station of LNs to be absent, but dissection was ruled out to be noncompliant if more than one station of LNs were not detected¹¹. In conclusion, LN dissection noncompliance is considered an effective method for assessing the quality of LN dissection.

Reference

1. Chen QY, Lin GT, Zhong Q, et al. Laparoscopic total gastrectomy for upper-middle advanced gastric cancer: analysis based on lymph node noncompliance. *Gastric Cancer*. 2020 Jan;23(1):184-194.
2. Chen QY, Zhong Q, Liu ZY, et al. Does Noncompliance in Lymph Node Dissection Affect Oncological Efficacy in Gastric Cancer Patients Undergoing Radical Gastrectomy? *Ann Surg Oncol*. 2019 Jun;26(6):1759-1771.
3. Kim MG, Kim KC, Kim BS, et al A totally laparoscopic distal gastrectomy can be an effective way of performing laparoscopic gastrectomy in obese patients (body mass index ≥ 30). *World J Surg*. 2011;35(6):1327-32.
4. Miyaki A, Imamura K, Kobayashi R, et al. Impact of visceral fat on laparoscopy-assisted distal gastrectomy. *Surgeon*. 2013;11(2):76-81.
5. Sugimoto M, Kinoshita T, Shibasaki H, et al Short-term outcome of total laparoscopic distal gastrectomy for overweight and obese patients with gastric cancer. *Surg Endosc*. 2013;27(11):4291-6.
6. Lee HJ, Kim HH, Kim MC, et al The impact of a high body mass index on laparoscopy assisted gastrectomy for gastric cancer. *Surg Endosc Other Intervent Tech*. 2009;23(11):2473-9.
7. Ojima T, Iwahashi M, Nakamori M, et al The impact of abdominal shape index of patients on laparoscopy-assisted distal gastrectomy for early gastric cancer. *Langenbecks Arch Surg*. 2012;397(3):437-45.
8. Lin GT, Chen QY, Zheng CH, et al. Lymph Node Noncompliance Affects the Long-Term Prognosis of Patients with Gastric Cancer after Laparoscopic Total Gastrectomy. *J Gastrointest Surg*. 2020 Mar;24(3):540-550.

9. Wagner PK, Ramaswamy A, Rüschoff J, et al. Lymph node counts in the upper abdomen: Anatomical basis for lymphadenectomy in gastric cancer. *Br J Surg.* 2010;78:825-827.
10. Maruyama K, Okabayashi K, Kinoshita T. Progress in gastric cancer surgery in Japan and its limits of radicality. *World J Surg.* 1987;11:418-425
11. De Steur WO, Hartgrink HH, Dikken JL, et al. Quality control of lymph node dissection in the Dutch Gastric Cancer Trial. *Br J Surg.* 2015;102:1388-1393.

11. Line 205 - 18F-FDG pet? please specify

The authors' answer: Thank you for your valuable comment. The Positron Emission Tomography (PET) referred to in this study is the 18F-fluorodeoxyglucose positron emission tomography/computed tomography (18F-FDG PET/CT), an imaging technique used to diagnose cancers and other diseases by detecting highly metabolically active tissues. We have made revisions to the *Method* of the revised manuscript.

Reference

1. Howard BA, Wong TZ. 18F-FDG-PET/CT Imaging for Gastrointestinal Malignancies. *Radiol Clin North Am.* 2021 Sep;59(5):737-753.

12. What was the intraoperative imaging protocol? Was FLU imaging performed on the fly? Or was it performed after clearing out each respective station to assess for any remaining LNs etc.?

The authors' answer: Thank you for your valuable comment. As stated in Question 8, during the surgical procedure, the surgeons tended to utilize the green fluorescence imaging mode to perform LN dissection. If necessary, they switched between white light and green fluorescence imaging modes to observe the surgical area. Following the LN dissection in each area, we also employed fluorescence imaging to assess the completeness of the LN dissection. We have added these contents to the *Method* of the revised manuscript.

Results

13. Line 241-what is meant with ICG contamination? How was / is contamination determined?

The authors' answer: Thank you for your valuable comment. ICG contamination occurs when fluorescent dyes drain indirectly through the lymphatic vessels and penetrate the gastric serosa layer because of failed injection during endoscopy. This leads to direct contamination of the adjacent peritoneal tissue (or even enter the adjacent omental bursa), causing non-specific staining in the field of view. Hence, non-LNs/vessels can also produce visualization under fluorescence mode, thereby interfering with the accurate visualization of normal LNs.

14. Line 248 - do the authors think that the tumor location is / was of effect on the LN yield in each group?

The authors' answer: Thank you for your valuable comment. The mean number of LNs retrieved from the ICG group was significantly higher than that retrieved from the non-ICG group (mean [SD], 50.5 [15.9] vs. 42.0 [10.3], respectively. Upon further subgroup analysis based on tumor location, we discovered that regardless of tumor location, the number of LNs retrieved in the ICG group was higher than that of the non-ICG group (lower: mean [SD], 50.0 [16.6] vs. 40.8 [9.9], $P=0.001$; upper/middle: mean [SD], 51.1 [14.9] vs. 42.8 [10.4]). This indicates that tumor location does not affect the LN yield for each group (ICG and non-ICG, $P=0.001$) (**eTable 1**).

eTable 1 The Relationship Between Tumor Location and Total Lymph Node Retrieved in ICG and non-ICG Groups

Characteristic	Mean (SD)		P Value
	ICG	non-ICG	

Upper/Middle	51.1 (14.9)	42.8 (10.4)	0.001
Lower	50.0 (16.6)	40.8 (9.9)	0.001

15. Line 249 and line 258 and line 272-despite the LN yield being much higher in the ICG group, adjuvant chemo rates were similar between the groups. Though the OS and DFS is better in the ICG group. Does this mean that adjuvant chemo has no effect on the treated patients, but the number of nodes harvested does? Could all be explained by the fact that in a fair number of subjects (12%) in the control arm fewer than 30 nodes were harvested (line 295)?

The authors' answer: Thank you for your comments. Our results (**Table 3**) revealed that adjuvant chemotherapy is an independent protective factor for OS and DFS in patients with GC ($P < 0.05$ for both). Postoperative adjuvant chemotherapy has been demonstrated to significantly improve the survival of patients with locally advanced GC¹⁻³, and the results of our study are also consistent with those of previous studies. Moreover, the multivariable Cox regression analysis also revealed ICG as an independent protective factor affecting OS and DFS, owing to the increased number of LNs retrieved in the ICG group and the significant reduction in LN dissection noncompliance⁴⁻⁷. Additionally, according to the *Japanese Gastric Cancer Treatment Guidelines*⁸, whether patients who underwent surgery receive chemotherapy mainly depends on postoperative pathological staging, and in our study, the postoperative AJCC staging data of the two groups of patients were comparable ($P = 0.429$). Thus, the proportion of patients receiving adjuvant chemotherapy in the two groups was comparable ($P = 0.210$).

Reference

1. Ohtsu A, Shimada Y, Shirao K, et al Randomized phase III trial of fluorouracil alone versus fluorouracil plus cisplatin versus uracil and tegafur plus mitomycin in patients with unresectable, advanced gastric cancer: the

- Japan Clinical Oncology Group Study (JCOG9205). *J Clin Oncol*. 2003;21:54–9.
2. Bang YJ, Van Cutsem E, Feyereislova A, et al Trastuzumab in combination with chemotherapy versus chemotherapy alone for treatment of HER2-positive advanced gastric or gastro-oesophageal junction cancer (ToGA): a phase 3, open-label, randomised controlled trial. *Lancet*. 2010;376:687–97.
 3. Glimelius B, Hofman K, Haglund U, et al Initial or delayed chemotherapy with best supportive care in advanced gastric cancer. *Ann Oncol*. 1994;5:189–90
 4. Smith DD, Schwarz RR, Schwarz RE. Impact of total lymph node count on staging and survival after gastrectomy for gastric cancer: data from a large US-population database. *Journal of clinical oncology : official journal of the American Society of Clinical Oncology*. 2005;23(28):7114-7124.
 5. Son T, Hyung WJ, Lee JH, et al. Clinical implication of an insufficient number of examined lymph nodes after curative resection for gastric cancer. *Cancer*. 2012;118(19):4687-4693.
 6. Huang CM, Lin JX, Zheng CH, et al. Prognostic impact of dissected lymph node count on patients with node-negative gastric cancer. *World journal of gastroenterology*. 2009;15(31):3926-3930.
 7. Seevaratnam R, Bocicariu A, Cardoso R, et al. A meta-analysis of D1 versus D2 lymph node dissection. *Gastric cancer: official journal of the International Gastric Cancer Association and the Japanese Gastric Cancer Association*. 2012;15 Suppl 1:S60-69
 8. Japanese Gastric Cancer Association. Japanese gastric cancer treatment guidelines 2018 (5th edition). *Gastric Cancer*. 2021 Jan;24(1):1-21.

16. Should like 302 be "less than 30 retrieved LNs?"

The authors' answer: Thank you for your comments. We apologize for any confusion caused and have revised the sentence to ensure that readers can easily comprehend it. "Further analysis of patients with ≥ 30 retrieved LNs revealed that the OS in the ICG group was significantly higher than that in the non-ICG group (log-rank $P=0.047$; 3-year OS rate: 86.0% vs. 76.1%) (eFigure 5)"

17. 30+% of non-compliance generally speakings seems really high to me.

Can the authors elaborate on how noncompliance occurs? Is it because its hard to remove tissue here? Or is it to risky to remove tissue from certain stations? Or is it that if visually no nodes are seen nothing will be removed?

The authors' answer: Thank you for your valuable comment. The LN dissection noncompliance in GC surgery has gradually been recognized by multiple RCT studies and applied for evaluating the quality of LN dissection¹⁻². LN dissection noncompliance is influenced by patient clinical status, tumor growth extent, surgeon-preferred strategies, and the pathologist's rigor in evaluating the excised specimens, which explains why some studies report different rates of LN dissection noncompliance for radical gastrectomy, ranging from 43.2% to 84%¹⁻⁵. Even with a standardized and systematic surgical procedure for GC LN dissection, missing microscopically involved LNs is possible. The reasons for LN dissection noncompliance are commonly related to several factors. First, in laparoscopic surgery, if the gastric tumor is large, it can compress the surgical field and affect the exposure of the stomach vessels and surrounding area of the lymphatic adipose tissue, which may result in missed LNs in this surgical area⁶. Second, different surgical techniques and adhesions may hinder the complete removal of LNs⁷. Third, numerous studies have reported that high BMI or increased intra-abdominal fat directly reduces the number of detected LNs⁸⁻¹². Patients with high BMI often have a large amount of adipose tissue accumulated in the abdomen, making it difficult to distinguish the relationship among pancreatic tissue, adipose tissue, and LNs during surgery, which therefore makes LN dissection more difficult. Especially for patients with high BMI, specific areas, such as the celiac trunk and pancreatic head, have very deep LNs, which further increases the difficulty of accurate positioning and dissection^{6,13}. Finally, the number of LNs considerably varies in each LN station in the normal gastric drainage LN area. Sometimes, some small stations (such as nos. 2, No.5, and No.10) may not contain any LNs at all, indicating biological variability¹⁴⁻¹⁵. Therefore, even if

surgeons perform LN dissection for all LN stations according to the D2 criteria for radical resection of GC, there may be no LNs in the specimen. Considering this biological variability, the Dutch Gastric Cancer Trial study considered that LN dissection for GC allowed one station of LNs to be absent, but dissection was ruled out to be noncompliant if more than one station of LNs were not detected¹⁶. This biological variation may also be another important reason for LN dissection noncompliance¹³.

Reference

1. Bonenkamp JJ, Hermans J, Sasako M, et al. Quality control of lymph node dissection in the Dutch randomized trial of D1 and D2 lymph node dissection for gastric cancer. *Gastric Cancer*. 1998;1:152–159.
2. Park YK, Yoon HM, Kim YW, et al. Laparoscopy-assisted versus Open D2 Distal Gastrectomy for Advanced Gastric Cancer: Results From a Randomized Phase II Multicenter Clinical Trial (COACT 1001). *Ann Surg*. 2018;267:638-645.
3. Bonenkamp JJ, Songun I, Hermans J, et al. Randomised comparison of morbidity after D1 and D2 dissection for gastric cancer in 996 Dutch patients. *Lancet*. 1995;345(8952): 745-748.
4. Bonenkamp JJ, Hermans J, Sasako M, et al. Extended Lymph-Node Dissection for Gastric Cancer. *N Engl J Med*. 1999;340:908–914.
5. Claassen YHM, De Steur WO, Hartgrink HH, et al. Surgicopathological quality control and protocol adherence to lymphadenectomy in the critics gastric cancer trial. *Ann Surg*. 2018;268:1008-1013
6. Chen QY, Lin GT, Zhong Q, et al. Laparoscopic total gastrectomy for upper-middle advanced gastric cancer: analysis based on lymph node noncompliance. *Gastric Cancer*. 2020 Jan;23(1):184-194.
7. Chen QY, Zhong Q, Liu ZY, et al. Does Noncompliance in Lymph Node Dissection Affect Oncological Efficacy in Gastric Cancer Patients Undergoing Radical Gastrectomy? *Ann Surg Oncol*. 2019 Jun;26(6):1759-1771.
8. Kim MG, Kim KC, Kim BS, et al. A totally laparoscopic distal gastrectomy can be an effective way of performing laparoscopic gastrectomy in obese patients (body mass index ≥ 30). *World J Surg*. 2011;35(6):1327-32.
9. Miyaki A, Imamura K, Kobayashi R, et al. Impact of visceral fat on laparoscopy-assisted distal gastrectomy. *Surgeon*. 2013;11(2):76-81.
10. Sugimoto M, Kinoshita T, Shibasaki H, et al. Short-term outcome of total laparoscopic distal gastrectomy for overweight and obese patients with gastric cancer. *Surg Endosc*. 2013;27(11):4291-6.
11. Lee HJ, Kim HH, Kim MC, et al The impact of a high body mass index on laparoscopy assisted gastrectomy for gastric cancer. *Surg Endosc Other*

- Intervent Tech. 2009;23(11):2473-9.
12. Ojima T, Iwahashi M, Nakamori M, et al The impact of abdominal shape index of patients on laparoscopy-assisted distal gastrectomy for early gastric cancer. *Langenbecks Arch Surg.* 2012;397(3):437-45.
 13. Lin GT, Chen QY, Zheng CH, et al. Lymph Node Noncompliance Affects the Long-Term Prognosis of Patients with Gastric Cancer after Laparoscopic Total Gastrectomy. *J Gastrointest Surg.* 2020 Mar;24(3):540-550.
 14. Wagner PK, Ramaswamy A, Rüschoff J, et al. Lymph node counts in the upper abdomen: Anatomical basis for lymphadenectomy in gastric cancer. *Br J Surg.* 2010;78:825-827.
 15. Maruyama K, Okabayashi K, Kinoshita T. Progress in gastric cancer surgery in Japan and its limits of radicality. *World J Surg.* 1987;11:418-425
 16. de Steur WO, Hartgrink HH, Dikken JL, et al. Quality control of lymph node dissection in the Dutch Gastric Cancer Trial. *Br J Surg.* 2015;102:1388-1393

18. There seems to be no significant differences in OS between compliant and non-compliant subjects, only in DFS, but only in the pN+ patients, although recurrence rates in these subjects were not significantly different between the compliant and non-compliant subjects. Though in the N0 groups OS and DFS rates are significantly different. How do the authors explain this?

The authors' answer: Thank you for your valuable comment. We apologize for any confusion caused. As mentioned by you, no significant differences in OS and DFS were observed between the patients who underwent compliant and noncompliant lymphadenectomy among all the patients (**eFigure 6**). However, further subgroup analysis revealed no significant differences in OS and DFS between the patients who underwent compliant and noncompliant lymphadenectomy among pN0 patients. By contrast, pN+ patients who underwent compliant lymphadenectomy had better 3-year DFS rates than those who underwent noncompliant lymphadenectomy (**eFigure 7**). Patients who underwent compliant lymphadenectomy had a 3-year cumulative recurrence rate of 34.8%, which was lower than the 50% rate in patients who

underwent noncompliant lymphadenectomy, although this difference was not statistically significant. Previous studies on noncompliant lymphadenectomy have also reported that it is an independent risk factor for long-term survival. This is because LNs are the main route of metastasis in patients with GC. Thus, D2 LN dissection should remove all possible LNs to effectively prevent potential (micro) metastases and improve prognosis. Our study also demonstrated that the rate of noncompliant lymphadenectomy was significantly higher in patients with local recurrence than that in those without it. This suggests that noncompliant lymphadenectomy increases the risk of recurrence to some extent in pN+ patients. We have added these contents in the *Discussion* of revised manuscript.

Reference

1. Chen QY, Zhong Q, Liu ZY, et al. Does Noncompliance in Lymph Node Dissection Affect Oncological Efficacy in Gastric Cancer Patients Undergoing Radical Gastrectomy? *Ann Surg Oncol*. 2019 Jun;26(6):1759-1771.
2. Chen QY, Lin GT, Zhong Q, et al. Laparoscopic total gastrectomy for upper-middle advanced gastric cancer: analysis based on lymph node noncompliance. *Gastric Cancer*. 2020 Jan;23(1):184-194.
3. Lin GT, Chen QY, Zheng CH, et al. Lymph Node Noncompliance Affects the Long-Term Prognosis of Patients with Gastric Cancer after Laparoscopic Total Gastrectomy. *J Gastrointest Surg*. 2020 Mar;24(3):540-550.
4. De Steur WO, Hartgrink HH, Dikken JL, et al. Quality control of lymph node dissection in the Dutch Gastric Cancer Trial. *British Journal of Surgery*. 2015;102(11):1388-1393.

Discussion:

19. Line 376 - this is new and comes out of the blue, particularly also because nothing about the H&E staining protocol is described. Please include the pathology assessment protocols into the methods section. Are nodes bisected? Are nodes serially sectioned and then evaluated? Etc. The more levels are collected for any given node the higher the

likelihood of finding (micro-) mets, however it comes at the cost of time.

The authors' answer: Thank you for your valuable comment. We have added these contents regarding the pathological evaluation protocol in the *Method* of our revised manuscript. After resecting the specimens, the surgeons positioned each LN station according to the location of the blood vessel clips retained in the specimens during the operation and sorted each LN station according to the *Japanese Research Society for Gastric Carcinoma criteria*¹. Two surgeons (J.-W.X. and Q.Z.) examined all specimens. The specimens were immediately sent to the Department of Pathology after repacking, and the LNs of each station were examined by two or more experienced pathologists by palpation and microscopy. All pathological examinations were performed in a standard manner²⁻⁵.

LNs containing isolated tumor cells, defined as single tumor cells or small clusters of cells ≤ 0.2 mm in greatest diameter, without stromal reaction, are classified as pN0 in GC⁵. There is no micro-metastasis (N1mi) category in staging GC⁶. LNs containing clusters of cells >0.2 mm in diameter are considered positive. In pretreated GCs, positive LNs are defined as having at least one focus of residual tumor cells in the LNs regardless of size. LNs with acellular mucin pool or fibrotic LNs with no viable tumor are considered negative⁶. In this study, all LNs were bisected and evaluated without routine serial sectioning. Information regarding H&E staining is provided in **Supplemental Digital Content 3** and has been incorporated into the *Method* of the revised manuscript.

Reference

1. Japanese Gastric Cancer Association. Japanese classification of gastric carcinoma: 3rd English edition. *Gastric Cancer*. 2011 Jun;14(2):101-12.
2. Chen QY, Xie JW, Zhong Q, et al. Safety and Efficacy of Indocyanine Green Tracer-Guided Lymph Node Dissection During Laparoscopic Radical Gastrectomy in Patients With Gastric Cancer: A Randomized Clinical Trial. *JAMA Surg* 2020; 155(4):300-311.
3. Chen QY, Lin GT, Zhong Q, et al. Laparoscopic total gastrectomy for

upper-middle advanced gastric cancer: analysis based on lymph node noncompliance. *Gastric Cancer*. 2020 Jan;23(1):184-194.

4. Lin GT, Chen QY, Zheng CH, et al. Lymph Node Noncompliance Affects the Long-Term Prognosis of Patients with Gastric Cancer after Laparoscopic Total Gastrectomy. *J Gastrointest Surg*. 2020 Mar;24(3):540-550.

5. International Collaboration on Cancer Reporting. *GASTRIC CANCER STRUCTURED REPORTING PROTOCOL: 2nd Edition*, 2020.

www.ICCR-Cancer.org

6. Amin MB, Edge SB, Greene FL, Byrd DR, Brookland RK, Washington MK, Gershenwald JE, Compton CC, Hess KR, Sullivan DC, Jessup JM, Brierley JD, Gaspar LE, Schilsky RL, Balch CM, Winchester DP, Asare EA, Madera M, Gress DM and Meyer LR (eds) (2017). *AJCC Cancer Staging Manual*. 8th Edition, Springer, New York

20. In following my comment of line 376 - have the authors considered triaging ICG+ LNs based on their fluorescence intensity to be able to predict those nodes that are at risk of harboring mets? Albeit slightly different Nishio et al., in Nature Comm 2019 described such an approach for an antibody-IRDye800.

The authors' answer: Thank you for your comments. As mentioned by the reviewer, predicting LN metastasis rapidly, in situ, and in vivo during surgery, in addition to pathological diagnosis, is currently a challenging research topic. However, ICG has limited diagnostic value for metastatic LNs as it is not a cancer-specific tracer¹⁻², which is the main disadvantage of ICG fluorescence imaging. Similar to the application of antibody-IRDye800, no relevant literature has reported on the application of ICG in GC. Meanwhile, we have also observed similar efforts and attempts, such as the development of RGD-modified distearyl acylphosphatidyl ethanolamine-polyethylene glycol micelle (DSPE-PEG-RGD) by Shao et al.³, which encapsulates ICG, accumulates better, and has longer circulation time in tumors. As described in Question 8, we typically used the green fluorescence mode for surgery and did not collect data on the fluorescence intensity of each group of LNs. Thus, further analysis of the relationship between fluorescence intensity and LN

metastasis is currently not feasible. We look forward to future research on intraoperative LN metastasis prediction using techniques, such as fluorescence intensity and specific antibody-labeled fluorescent dyes. We have added the above descriptions to the limitations of *Discussion*.

Reference

1. Yano K, Nimura H, Mitsumori N, et al. The efficiency of micrometastasis by sentinel node navigation surgery using indocyanine green and infrared ray laparoscopy system for gastric cancer. *Gastric Cancer* (2012) 15(3):287-91.
2. Villegas-Tovar E, Jimenez-Lillo J, Jimenez-Valerio V, et al. Performance of indocyanine green for sentinel lymph node mapping and lymph node metastasis in colorectal cancer: A diagnostic test accuracy meta-analysis. *Surg Endosc* (2020) 34(3):1035-47.
3. Shao J, Zheng X, Feng L, et al. Targeting fluorescence imaging of RGD-modified indocyanine green micelles on gastric cancer. *Front Bioeng Biotechnol* (2020), 8:575365.

21. Line 408-ish - what about in-transit LN mets?

The authors' answer: Thank you for your comments. Based on your suggestion, we have revised lines 408-409 to read, "improper manipulation of lymphatic adipose tissue often leads to the release of free cancer cells from the lymphovascular pedicle and metastatic LN, thereby increasing the risk of recurrence."

Figures/Tables

22. Did the authors look at the relationship between BMI and noncompliance in the ICG-group?

The authors' answer: Thank you for your comments. As per your request, we divided the ICG group patients into three categories based on their BMI: BMI<24, 24≤BMI<28, and BMI≥28¹. We found no significant difference in noncompliance and compliance between different BMI categories in the ICG group patients ($P=0.627$) (**eTable 2**).

eTable 2 The Relationship Between BMI and Lymph Node Dissection Noncompliance in the ICG Group

Characteristic	No. (%)			P Value
	BMI<24	24≤BMI<28	BMI≥28	0.627
Noncompliance	24 (31.6)	13 (29.5)	4 (44.4)	
Compliance	52 (68.4)	31 (70.5)	5 (55.6)	

BMI, body mass index

Reference

1.Zhou BF; Cooperative Meta-Analysis Group of the Working Group on Obesity in China. Predictive values of body mass index and waist circumference for risk factors of certain related diseases in Chinese adults--study on optimal cut-off points of body mass index and waist circumference in Chinese adults. *Biomed Environ Sci.* 2002 Mar;15(1):83-96.

General questions:

23. ICG-based LN harvesting is a rather universal technology and has successfully been implemented during both open, and (robot-assisted laparoscopic procedures. Why were only subjects that were operated laparoscopically included?

The authors' answer: Thank you for your comments. The recent therapeutic effectiveness of robot-assisted gastrectomy guided by indocyanine green (ICG) has been reported¹⁻⁴. Woo Jin Hyung et al. have reported that the use of ICG in robots can help identify and retrieve all necessary LNs during surgery, enabling complete and thorough LN dissection¹. While the application of ICG in open gastrectomy is focused on early-stage GC sentinel LNs research⁵⁻⁶. Yang et al. have reported on the use of ICG in tracing the gastric lymphatic network of advanced GC and concluded that ICG is not suitable for patients with advanced GC who undergo open surgery⁷. Moreover, the short-term and long-term efficacy of laparoscopic surgery has been shown to be non-inferior to that of open surgery⁸⁻¹⁰. However, whether the oncological effectiveness of robot-assisted gastrectomy is non-inferior to that of laparoscopic gastrectomy remains unclear and should be analyzed in future large-sample randomized

controlled trials. Therefore, this study enrolled patients with GC who underwent laparoscopic surgery. In the future, we also hope to further develop research on robot-assisted ICG, and we have supplemented the above contents in the *Discussion* section of the revised manuscript.

Reference

- 1.Kwon IG, Son T, Kim HI, et al. Fluorescent Lymphography-Guided Lymphadenectomy During Robotic Radical Gastrectomy for Gastric Cancer. *JAMA Surg.* 2019 Feb 1;154(2):150-158.
- 2.Herrera-Almario G, Patane M, Sarkaria I, et al. Initial report of near-infrared fluorescence imaging as an intraoperative adjunct for lymph node harvesting during robot-assisted laparoscopic gastrectomy. *J Surg Oncol.* 2016;113(7):768-770.
- 3.Osterkamp J, Strandby R, Nerup N, et al. Intraoperative near-infrared lymphography with indocyanine green may aid lymph node dissection during robot-assisted resection of gastroesophageal junction cancer. *Surg Endosc.* 2023 Mar;37(3):1985-1993.
4. Romanzi A, Mancini R, Ioni L, et al. ICG-NIR-guided lymph node dissection during robotic subtotal gastrectomy for gastric cancer. A single-centre experience. *Int J Med Robot* (2021) 17(2):e2213.
- 5.Tajima Y, Yamazaki K, Masuda Y, et al. Sentinel node mapping guided by indocyanine green fluorescence imaging in gastric cancer. *Ann Surg.* 2009 Jan;249(1):58-62.
- 6.Tajima Y, Murakami M, Yamazaki K, et al. Sentinel node mapping guided by indocyanine green fluorescence imaging during laparoscopic surgery in gastric cancer. *Ann Surg Oncol.* 2010 Jul;17(7):1787-93.
- 7.Park JH, Berlth F, Wang C, et al. Mapping of the perigastric lymphatic network using indocyanine green fluorescence imaging and tissue marking dye in clinically advanced gastric cancer. *Eur J Surg Oncol.* 2022 Feb;48(2):411-417.
- 8.Yu J, Huang C, Sun Y, et al; Chinese Laparoscopic Gastrointestinal Surgery Study (CLASS) Group. Effect of Laparoscopic vs Open Distal Gastrectomy on 3-Year Disease-Free Survival in Patients With Locally Advanced Gastric Cancer: The CLASS-01 Randomized Clinical Trial. *JAMA.* 2019 May 28;321(20):1983-1992.
- 9.Son SY, Hur H, Hyung WJ, et al; Korean Laparoendoscopic Gastrointestinal Surgery Study (KLASS) Group. Laparoscopic vs Open Distal Gastrectomy for Locally Advanced Gastric Cancer: 5-Year Outcomes of the KLASS-02 Randomized Clinical Trial. *JAMA Surg.* 2022 Oct 1;157(10):879-886.
- 10.Etoh T, Ohyama T, Sakuramoto S, et al; Japanese Laparoscopic Surgery Study Group (JLSSG). Five-Year Survival Outcomes of Laparoscopy-Assisted

vs Open Distal Gastrectomy for Advanced Gastric Cancer: The JLSSG0901 Randomized Clinical Trial. JAMA Surg. 2023 Mar 15:e230096.

24. How many surgeons were involved in this study? Were the surgeons all experienced with gastrectomy and LN mapping. Were they all experienced in the use of ICG for LN mapping? If not, did you find any differences in LN yield between naive and experiences surgeons with either technology?

The authors' answer: Thank you for your valuable comment. The surgical team involved in this study was from the Department of Gastric Cancer at Fujian Medical University Union Hospital, comprising two surgeons (C.-H.Z. and C.-M.H.). All the participating surgeons in our study met the following criteria: they had performed more than 100 laparoscopic radical gastrectomies, completed a learning curve in laparoscopic radical LN dissection, passed the blind surgical video examination, and had ample experience in ICG-guided LN dissection for GC. We have added these details to the *Method* in the revised manuscript.

Reviewer #2 - Biostatistics, Clinical trials (Remarks to the Author):

This article reports the long-term outcomes from a study in gastric cancer patients randomized to receive either ICG fluorescence imaging vs non-ICG. The primary endpoint of the original study was the number of lymph nodes retrieved and this analysis has been already published. The current article now presents the secondary objectives of the original study: survival outcomes and recurrence patterns for these patients. The paper should state earlier and clearer that the reported analyses are based on the original randomized study.

The authors' answer: Thank you for your comments. We have clarified that this study is the secondary outcome report based on the original study¹ in the

Introduction of the revised manuscript.

Reference

1. Chen QY, Xie JW, Zhong Q, et al. Safety and Efficacy of Indocyanine Green Tracer-Guided Lymph Node Dissection During Laparoscopic Radical Gastrectomy in Patients With Gastric Cancer: A Randomized Clinical Trial. *JAMA surgery*. 2020.

There are a lot of subset analyses reported in this paper that do not appear to have been prespecified in the original protocol. This not only increases the number of tests performed and thus increases the false discovery rate, but also could be questioned as data driven.

The authors' answer: Thank you for your comments. The primary endpoint of our study (the number of LNs retrieved) has been reported previously¹. However, this investigation focuses on the protocol-based secondary endpoints (3-year DFS and OS), and our analysis revealed that the ICG group exhibited superior survival outcomes. Accordingly, our conclusion is based on this finding. As no literature has reported on this outcome previously, in an attempt to explain the survival benefit for the ICG group, we performed subset analyses to explore the potential reasons, such as LN yield and LN dissection noncompliance. However, as noted by the reviewer, these results were not pre-specified in the original protocol and may increase the false discovery rate. Therefore, we did not draw any conclusions from these results. Similar to previous high-quality studies²⁻⁶, this research analyzes prospective trial datasets to some extent, which does not increase the patient's additional risk or healthcare expenditure while addressing clinical problems. This study demonstrates the maximum utilization of existing clinical research data. Although multi-center ICG studies are ongoing, prospective studies often have long cycles and high costs. Therefore, advanced treatment decisions can typically only be made in specialized institutions until reliable results from RCTs emerge. For example, in 1994, Kitano et al. have reported on the

application of LDG in early GC⁷, but it was not until the CLASS-01 study in 2019 that the oncological efficacy of LDG in advanced GC was confirmed to be no less than that of ODG⁸. The fact that high-quality research lags behind clinical practice often limits the rapid development of medicine worldwide, especially in surgical interventions⁹. Our study provides a theoretical basis and reference for subsequent multi-center studies.

Reference

1. Chen QY, Xie JW, Zhong Q, et al. Safety and Efficacy of Indocyanine Green Tracer-Guided Lymph Node Dissection During Laparoscopic Radical Gastrectomy in Patients With Gastric Cancer: A Randomized Clinical Trial. *JAMA Surg* 2020; 155(4):300-311.
2. Chen QY, Zhong Q, Liu ZY, et al. Surgical Outcomes, Technical Performance and Surgery Burden of Robotic Total Gastrectomy for Locally Advanced Gastric Cancer: A Prospective Study. *Ann Surg*. 2021 Jan 22
3. Liu ZY, Chen QY, Zhong Q, et al. Intraoperative Adverse Events, Technical Performance, and Surgical Outcomes in Laparoscopic Radical Surgery for Gastric Cancer: A Pooled Analysis from two Randomized Trials. *Ann Surg*. 2022 Oct 17.
4. Xu BB, Lu J, Zheng ZF, et al. The predictive value of the preoperative C-reactive protein-albumin ratio for early recurrence and chemotherapy benefit in patients with gastric cancer after radical gastrectomy: using randomized phase III trial data. *Gastric Cancer*. 2019 Sep;22(5):1016-1028
5. Park SH, Hyung WJ, Yang HK, et al. Standard follow-up after curative surgery for advanced gastric cancer: secondary analysis of a multicentre randomized clinical trial (KLASS-02). *Br J Surg*. 2023 Mar 30;110(4):449-455.
6. Song JH, Shin HJ, Hyung WJ, et al. Predictive Value of KLASS-02-QC Assessment Score on KLASS-02 surgical Outcomes: Validation of Surgeon Quality Control and Standardization for D2 Lymphadenectomy. *Ann Surg*. 2023 Jan 23.
7. Kitano S, Iso Y, Moriyama M, et al. Laparoscopy-assisted Billroth I gastrectomy. *Surg Laparosc Endosc* 1994; 4(2):146-8
8. Yu J, Huang C, Sun Y, et al. Effect of Laparoscopic vs Open Distal Gastrectomy on 3-Year Disease-Free Survival in Patients With Locally Advanced Gastric Cancer: The CLASS-01 Randomized Clinical Trial. *JAMA*. 2019 May 28;321(20):1983-1992.
9. Wang JB, Zhong Q, Chen QY, et al. Well-designed retrospective study versus small-sample prospective study in research based on laparoscopic and open radical distal gastrectomy for advanced gastric cancer. *Surg Endosc*. 2020 Oct;34(10):4504-4515.

It was not explained why the ITT analysis was not attempted: was peritoneal metastasis part of the exclusion criteria? Why were these patients excluded? Also, for the survival analyses, withdrawn from study patients could have been censored.

The authors' answer: Thank you for your comments. One of the withdrawal criteria in this study was peritoneal metastasis, as patients with peritoneal metastasis of GC cannot be cured via surgery and should be treated mainly with systematic chemotherapy; thus, they were not included in the study. Only one patient in the study was contaminated with ICG and did not undergo ICG surgery. Due to this small sample size alteration, an ITT analysis was not performed. In response to the reviewer's request, we performed the ITT analysis and supplemented the corresponding results of LNs retrieved (primary outcome) and 3-year survival (secondary outcome).

In the ITT analysis, there were 130 patients in the ICG group and 129 patients in the non-ICG group, with a mean (SD) total number of LNs retrieved of 50.6 (15.9) and 42.0 (10.3), respectively ($P<0.001$). Moreover, survival analysis revealed that the OS and DFS of the ICG group were both superior to those of the non-ICG group (log-rank $P<0.05$) (**eFigure 1**). These findings were consistent with the conclusions drawn from the PP analysis. Additionally, because the patients who withdrew from the study did not receive surgical treatment, we excluded them from the survival analysis. We have revised these supplement in the **Method** and **Results** of the manuscript.

Figure 1. Trial Profile

eFigure 1. ITT Analysis: Kaplan-Meier Curves Comparing Overall Survival (A) and Disease-free Survival (B) Between the ICG Group and Non-ICG Group

It is not specified for the multivariable analysis, what screening threshold was used in the univariable regressions to be included in the multivariable analyses.

The authors' answer: Thank you for your comments. Factors with a P -value < 0.05 in the univariate analysis will be included in further multivariate analysis. We have supplemented the contents in the *Method* of revised manuscript.

More details should be provided about the method (and reference for it) used for the competing risk analysis. There were 2 p values included in Table 2 and they have not been properly explained.

The authors' answer: Thank you for your comments. **Table 2** contains two kinds of P -values: the first one is the P -value for the Cox analysis of the hazard ratio, and the second one is the P -value for the chi-square test of different events (any recurrence, locoregional, or peritoneum) between the ICG and non-ICG groups. As per your suggestion, we have added details (**Supplemental Digital Content 3**) on the competing risk analysis and relevant references¹ in the *Methods* section of our revised manuscript. And a detailed description is provided in the caption of **Table 2**.

Reference

1. Yu J, Huang C, Sun Y, et al; Chinese Laparoscopic Gastrointestinal Surgery Study (CLASS) Group. Effect of Laparoscopic vs Open Distal Gastrectomy on 3-Year Disease-Free Survival in Patients With Locally Advanced Gastric Cancer: The CLASS-01 Randomized Clinical Trial. *JAMA*. 2019 May 28;321(20):1983-1992.

There were some comparisons made for the overall study population (combining the 2 arms: eFigures 4 and 6). This has not been justified. Furthermore, an interaction between ICG and LN compliance has been assessed among patients with pN+, however it would have been helpful to evaluate such interactions for other variables for which instead stratified analyses have been performed; for example, interaction between ICG arm and number of retrieved lymph nodes (continuous), between ICG arm and LN compliance in the full population, etc.

The authors' answer: Thank you for your comments. This study aims to report the secondary endpoints of the RCT study, namely the long-term oncological outcomes of ICG fluorescence-guided LN dissection in GC. The results revealed that compared to conventional surgery, ICG surgery significantly improved the 3-year prognosis of patients with GC. To justify the survival benefits of patients with ICG, various subgroup analyses performed for post hoc analysis based on prospective research. As the ICG group had no patients with less than 30 LNs retrieved, interactions related to "retrieved LNs (continuous)" could not be analyzed. Therefore, we used the stratification method of **eFigures 4** and **eFigures 6**. As per your suggestion, we analyzed the interaction between ICG grouping and LN dissection compliance among the full population, and no significant interactive effect of ICG and LN dissection compliance on OS and DFS was identified (P -interaction for OS= 0.077, P -interaction for DFS= 0.125). We have added this contents to the *Results* of the revised manuscript.

Discussion claims that "...for the same pT stage, the possibility of detecting LM metastasis in the ICG group was higher than that in the non-ICG group" (eFigure 3) however no p-value was provided and no method referenced.

The authors' answer: Thank you for your comments. We have added to the legend of eFigure 3 the *P* value (chi-square test) for pN staging between the ICG and non-ICG groups stratified by different pT stages.

eFigure 3. Distribution of pN Stages of Each pT Stage in the ICG Group and Non-ICG Group ($P_{pT1}=0.900$; $P_{pT2}=0.595$; $P_{pT3}=0.841$; $P_{pT4}=0.088$).

Some comparisons were highlighted when they had a significant p value, but some non-significant p values were not mentioned (for example eTable 3, OS between ICG vs non ICG among patients with 30 or more lymph nodes retrieved. There were only 16 patients that had less than 30 nodes retrieved so when comparing ICG vs non-ICG in the full population (significant pvalue) vs only among those with ≥ 30 nodes (pvalue is no longer statistically significant) it appears that these 16 patients with less than 30 nodes play a role.

The authors' answer: Thank you for your comments. According to your suggestion, we have emphasised non-significant *P* values in the *Results* of

revised manuscript. "Multivariate Cox regression analysis (**eTable 4**) showed that compared with conventional lymphadenectomy, ICG fluorescence imaging-guided lymphadenectomy was an independent protective factor for DFS in patients with ≥ 30 retrieved LNs (ICG vs. non-ICG, HR=0.52; 95%CI, 0.29-0.92; $P=0.024$). While uni- and multivariate analyses indicated that ICG fluorescence imaging-guided lymphadenectomy was not an independent factor for OS in patients with ≥ 30 retrieved LNs ($P>0.05$)."

Absolute differences and risk differences are reported however risk difference was defined only in the legend of a table. I would recommend removing those quantities or providing more details and reference.

The authors' answer: Thank you for your comments. As per your request, we have supplemented the *Method* with the more details and reference¹ for risk differences (**Supplemental Digital Content 3**).

Reference

1. Yu J, Huang C, Sun Y, et al; Chinese Laparoscopic Gastrointestinal Surgery Study (CLASS) Group. Effect of Laparoscopic vs Open Distal Gastrectomy on 3-Year Disease-Free Survival in Patients With Locally Advanced Gastric Cancer: The CLASS-01 Randomized Clinical Trial. *JAMA*. 2019 May 28;321(20):1983-1992.

The hazard ratios for the competing risk analysis were above 1 and that contradicts the interpretations in the paper. Has the reference group been switched?

The authors' answer: Thank you for your comments. In accordance with your suggestions, we have rechecked the hazard ratios for the competing risk analysis regarding the recurrence within 3 years in **Table 2** and recalculated them. The results are as follows. We have also made revisions to the *Results* section of the revised manuscript. A detailed description is provided in the

caption of **Table 2.**

Table 2. Frequencies of Causes of First Recurrence and Death **Within 3 Years** After Surgery in ICG and Non-ICG Groups

Events	Surgery, No. (%)		Risk Difference ^a	Hazard Ratio (95% CI) ^b	P Value ^c	P for chi-square test
	ICG Group (n = 129)	Non-ICG Group (n = 129)				
Any recurrence^d	23 (17.8)	40 (31.0)	-0.131	0.53 (0.32-0.89)	0.017	0.014
Locoregional	2 (1.6)	10 (7.8)	-0.073	0.19 (0.04-0.85)	0.030	0.018
Peritoneum	9 (7.0)	10 (7.8)	-0.009	0.87 (0.35-2.13)	0.752	0.812
Liver	2 (1.6)	6 (4.7)	-0.032	0.33 (0.07-1.62)	0.170	0.281
Multiple sites ^e	4 (3.1)	5(3.9)	-0.009	0.79 (0.21-2.92)	0.718	>0.999
Other or uncertain sites ^f	6 (4.7)	9 (7.0)	-0.026	0.64 (0.23-1.81)	0.402	0.425
Cause of death^g	18 (14.0)	34 (26.4)	-0.124	0.50 (0.28-0.89)	0.018	0.013
Gastric cancer	17 (94.4)	32 (94.1)	-0.116	0.50 (0.28-0.91)	0.022	>0.999
Other causes ^h	1 (5.6)	2 (5.9)	-0.011	0.47 (0.04-5.21)	0.540	

^a Except for all-cause death, the risk difference was calculated by subtracting the cumulative incidence in the first 3 years of the Non-ICG group from that of the ICG group, in presence of competing events; for all-cause death, the risk difference was calculated by subtracting the 3-year overall survival rate of the Non-ICG group from that of the ICG group.

^b Except for all-cause death, competing-risks survival regression was used to derive the hazard ratio, 95% CI, and P value. For total recurrence, all-cause death was the competing event; for the specific types of recurrence, other types of recurrence and death were the competing events; for gastric cancer cause of death, other causes of death were the competing events, and vice versa. Univariate Cox regression was used for all-cause death. **Non-ICG group is the reference group.**

^c P value for the hazard ratios.

^d Refers only to first-time recurrence, even though patients can have recurrence at multiple times.

^e Includes patients who have recurrence simultaneously in 2 or more metastatic sites, including peritoneum, liver, lung, bone, brain, distant lymph node, or other hematogenous metastatic sites.

^f Includes hematogenous recurrence at sites other than liver (ie, lung, bone, brain, adrenal gland), recurrence at distant lymph node, and recurrence at uncertain sites.

^g Post hoc exploratory outcomes.

^h Includes other cancers, diseases other than cancer, unintentional injuries, and unknown causes.

Cumulative incidence of locoregional recurrence is listed but no timepoint provided in the paper at which they were evaluated.

The authors' answer: Thank you for your comments. **Table 2** displays the incidence of recurrence during the 36 months postoperatively. **eTable 5** displays clinical characteristics of each patient who occurred locoregional recurrence **within 3 years**. For clarity, we have changed the table titles of **Table 2** and **eTable 5**, and the corresponding descriptions in the *Results* of the revised manuscript.

The log-rank test compares the overall curves not a specific timepoint as implied in some parts (for example in Abstract).

The authors' answer: Thank you for your comments. We apologize for any misunderstandings caused by errors in our writing. We have made corrections to the *Abstract* and *Results* sections of the revised manuscript accordingly. In the *Abstract*, we have revised the statement to read as follows: "Both OS and DFS in the ICG group were significantly higher than those in the non-ICG group (log-rank $P=0.015$; log-rank $P=0.012$). The 3-year OS rates in the ICG and non-ICG groups were 86.0% and 73.6%, respectively, with an absolute risk difference of 12.4%. The 3-year DFS rates in the ICG and non-ICG groups were 81.4% and 68.2%, respectively, with an absolute risk difference of 13.2%".

To the editors

1. Following the requirements of *Nature Communications*, we have restructured the abstract.

2. The description of Highlights has been deleted.
 3. In accordance with *Nature Communications's* requirements, we have rearranged the article structure, with *Method* placed after *Discussion*.
 4. To avoid using language such as "first" or "extremely," we have modified the wording of the article to make it more scientific.
- Best wishes.

Reviewers' Comments:

Reviewer #2:

Remarks to the Author:

Thank you for addressing the comments that I raised. Responses to my questions are very clear and I appreciate the authors including the additionally provided information into the manuscript.

Reviewer #3:

Remarks to the Author:

There are still multiple concerns about the paper in the current form. There are numerous post-hoc subset analyses performed and the authors failed to acknowledge that in the Discussion. The results from some of these analyses are overstated and an explanation not always included. Some of the analyses are univariate analyses and some are multivariable without a justification why multivariable analyses were not performed (for example the competing risk analysis, interaction analyses, etc). This is particularly important given that some results are significant in the univariable analyses however no longer significant in the presence of other factors. In the absence of a significant p-value the results are interpreted as "comparable" however, failure to reject the null hypothesis does not imply acceptance of the null hypothesis. Some analyses combine the two arms without a rationale for doing it. Among other, results for eFig 3 are overstated given the p-values are not significant.

Some of the tests performed are incorrect such as the chi-square test for the recurrence endpoints. These endpoints are time to event endpoints and should not be handled as categorical outcomes.

References for competing risk analysis and risk difference were requested and the reference provided is of another paper employing the same type of analysis rather than the reference for the method employed.

There was no rationale on how the cutpoint of 30 for lymph nodes.

eFig 3 is confusing, please clarify.

RESPONSE TO REVIEWERS

Title: **“Long-Term Outcomes of Indocyanine Green Fluorescence Imaging-Guided versus Conventional Laparoscopic Lymphadenectomy for Gastric Cancer”** We are extremely grateful to the editor and anonymous reviewer for their valuable comments and suggestions, which have helped improve the quality of the manuscript. Simultaneously, we would like to thank the reviewers and editors again for taking the time to review our manuscript. We would be very appreciative of your kind consideration of this paper.

After careful consideration of the reviewers' comments, we have revised the manuscript according to the referee's suggestions. Our descriptions of the revisions are as follows.

Reviewers' comments:

Reviewer #2 (Remarks to the Author):

Thank you for addressing the comments that I raised. Responses to my questions are very clear and I appreciate the authors including the additionally provided information into the manuscript.

The authors' answer: Thank you for your insightful feedback. We sincerely hope that this article can provide a high-level theoretical basis for the future benefits of ICG technology, promote the widespread of this technology, and benefit more patients with gastric cancer.

Reviewer #3 (Remarks to the Author):

There are still multiple concerns about the paper in the current form. There are numerous post-hoc subset analyses performed and the authors failed to acknowledge that in the Discussion. The results from some of these analyses

are overstated and an explanation not always included. Some of the analyses are univariate analyses and some are multivariable without a justification why multivariable analyses were not performed (for example the competing risk analysis, interaction analyses, etc). This is particularly important given that some results are significant in the univariable analyses however no longer significant in the presence of other factors. In the absence of a significant p-value the results are interpreted as "comparable" however, failure to reject the null hypothesis does not imply acceptance of the null hypothesis. Some analyses combine the two arms without a rationale for doing it. Among other, results for eFig 3 are overstated given the p-values are not significant.

The author's answer: Many thanks to the reviewer for pointing out these issues. We greatly appreciate the meticulousness of the reviewers in assessing the level of evidence. Consequently, we have revised the contents in discussion to address the inadequacies by including a detailed description of the post hoc analysis based on RCT. "In addition, this study adopted a post-hoc analysis based on RCT to attempt to explain the survival reasons for ICG patients' benefits. Therefore, caution should be exercised when promoting the conclusions of this study." . Furthermore, we have omitted the content and description pertaining to the results of **eFig.3** from the manuscript.

Regarding the reviewer's comment on the absence of multivariable analyses (such as competing risk analysis, interaction analyses, etc.), we were unable to perform a multivariable analysis due to the limitations of the competing risk analysis in **Table 2**. However, in response to the reviewer's request for interaction analyses, we have already addressed this issue in our first revision by providing additional information in **Lines 214-218**, as indicated by the reviewer and highlighted in red. As follows: "Since there were no patients in the ICG group with less than 30 LN dissections, interactions related to "retrieved LNs (continuous)" could not be analyzed. Therefore, we used the stratification method of **eFigures 4** and **eFigures 6**. According to your request, we analyzed the interaction between ICG grouping and LN dissection

compliance among the full population, and no significant interactive effect of ICG and LN compliance on OS and DFS (P -interaction for OS= 0.077, P -interaction for DFS= 0.125)". Simultaneously, in an effort to address any potential misunderstandings resulting from our unclear expression, we have rephrased the statement and highlighted it in red to make it more prominent.

To elucidate the underlying reasons for the survival advantages observed in ICG patients, we conducted post hoc analysis based on prospective research utilizing various subgroup analyses. Our objective was to explore the combinations of factors that could potentially impact patient prognosis and shed light on the potential reasons for the benefits observed in this patient population. Importantly, it should be emphasized that these analyses do not undermine the main conclusion of our study, which states that "Compared with conventional lymphadenectomy, ICG-guided laparoscopic lymphadenectomy is both safe and effective in prolonging the survival of patients with resectable GC."

Some of the tests performed are incorrect such as the chi-square test for the recurrence endpoints. These endpoints are time to event endpoints and should not be handled as categorical outcomes.

The author's answer: Thank you for your comments. As per the reviewer's request, we have removed the description of the results for the chi-square test for the recurrence endpoints in **Table 2**. This modification, however, does not affect the conclusions of the manuscript.

References for competing risk analysis and risk difference were requested and the reference provided is of another paper employing the same type of analysis rather than the reference for the method employed.

The author's answer: Thank you for your comments. The competing risk

model is a statistical method used to analyze survival data with multiple endpoints, and it was first introduced as a semi-parametric proportional hazards model for subdistribution by Fine and Gray in 1999¹. In accordance with the reviewer's request, we have included references regarding competing risk analysis and risk differences¹⁻⁵.

Reference

1. Fine JP, Gray RJ. A proportional hazards model for the subdistribution of a competing risk. *Journal of the American Statistical Association*. 1999; 94:496–509.
2. Beyersmann J, Latouche A, Buchholz A, Schumacher M. Simulating competing risks data in survival analysis. *Stat Med*. 2009 Mar 15;28(6):956-71.
3. Scheike TH, Zhang MJ. Analyzing Competing Risk Data Using the R *timereg* Package. *J Stat Softw*. 2011 Jan;38(2):i02.
4. Tripepi G, Jager KJ, Dekker FW, Wanner C, Zoccali C. Measures of effect: relative risks, odds ratios, risk difference, and 'number needed to treat'. *Kidney Int*. 2007 Oct;72(7):789-91.
5. Schechtman E. Odds ratio, relative risk, absolute risk reduction, and the number needed to treat--which of these should we use? *Value Health*. 2002 Sep-Oct;5(5):431-6.

There was no rationale on how the cutpoint of 30 for lymph nodes.

The author's answer: Thank you for your comments. In the method of the revised manuscript, we have explained the reasons behind using a cutpoint of 30 for lymph nodes. This decision is based on the description in the 8th edition of the AJCC (American Joint Committee on Cancer) staging for gastric cancer, which suggests "Although it is suggested that at least 16 regional nodes be removed/assessed pathologically, removal/evaluation of more nodes (≥ 30) is desirable"¹⁻². We apologize if this was not highlighted clearly in the previous revision, resulting in it being overlooked. We have now highlighted the relevant description in the manuscript.

Reference

1. Woo Y, Goldner B, Ituarte P, et al. Lymphadenectomy with Optimum of 29 Lymph Nodes Retrieved Associated with Improved Survival in Advanced

Gastric Cancer: A 25,000-Patient International Database Study. J Am Coll Surg. 2017 Apr;224(4):546-555.
2. Kakar S PT, Allen P. AJCC Cancer Staging Manual. 8th ed.: New York, NY: Springer-Verlag; 2017.

eFig 3 is confusing, please clarify.

The author's answer: Thank you for your comments. To try to explain the reason for the survival benefits of ICG patients, various subgroup analyses were used for post hoc analysis based on prospective research. Therefore, we performed the subgroup analysis by combining factors that may influence patient prognosis in order to explain the potential reasons for the benefits observed in this patient population. Importantly, these analyses do not affect the main conclusions of the article. In the revised manuscript, we have removed the content and description related to the results of **eFig 3**.

REVIEWER#4'S REPORT FOR FIRST SUBMISSION

Dear Authors,

I read the results of your RTC, Long-Term Outcomes of Indocyanine Green Fluorescence Imaging-Guided versus Conventional Laparoscopic Lymphadenectomy for Gastric Cancer, with pleasure.

The study was well-powered and appropriately designed to demonstrate the significant differences of your study results in nodal retrieval. The study results demonstrating improved D2 nodal dissection with the use of ICG for nodal identification and the associated improved long-term survival are an enormously important finding. Particularly supporting the importance of a through D2 dissection is reflected in the significant difference you showed in locoregional recurrence rates between your two cohorts. Your study will have

significant impact on the surgical practice for those treating patients with locally advanced gastric cancer.

I have several comments and questions related to the study:

1. While the study was not designed to compare between N+ and N- disease, it would be worth understanding whether or not IGC-guided lymphadenectomy differed in these two groups.

The author's answer: We will be happy to edit the text further, based on helpful comments from the reviewer. **eFigure 3** shows that for pN0 patients, there is no statistically significant difference in prognosis between ICG and non-ICG patients (OS: $P=0.083$; DFS: $P=0.083$). However, for pN+ patients, the prognosis of ICG patients is significantly better than that of non-ICG patients (OS: $P=0.023$; DFS: $P=0.012$).

eFigure 3. Kaplan-Meier Curves Comparing Overall Survival and Disease-free

Survival Between pN0 (A-B) and pN+ (C-D) .

2. Would also like to know in which nodal station the node retrieval numbers were increased?

The author's answer: Thank you for your comments. Our previous publication¹ (doi: 10.1001/jamasurg.2019.6033) demonstrated that for distal gastrectomy, the number of lymph node dissections in the ICG group was higher than that of the non-ICG group in the same lymph node station, especially in stations 4, 6, and 7. For total gastrectomy, the number of lymph node dissections in the ICG group was higher than that of the non-ICG group in the same lymph node station, especially in stations 4sa, 7, 11d, and 12a.

We divided the lymph nodes into perigastric regions (stations 1-6) and extraperigastric regions (stations 7-9, 11, and 12a). The number of perigastric and extraperigastric lymph node dissections in the ICG group was significantly higher than the number of dissections the non-ICG group. Moreover, the mean (SD) total number of retrieved lymph nodes in the ICG group was 49.6 (15.0) within the scope of D2 lymphadenectomy, which was significantly more than the total number of 41.7 (10.2) retrieved lymph nodes in the non-ICG group ($P < 0.001$).

Regardless of whether distal gastrectomy or total gastrectomy was performed, the ICG group underwent a significantly greater mean number of lymph node dissections than the non-ICG group based on the D2 criteria (for distal gastrectomy, mean [SD], 48.5 [13.7] vs 39.8 [10.1] dissections, respectively; $P < 0.001$; for total gastrectomy, mean [SD] 50.9 [16.3] vs 42.7 [10.2], respectively; $P = 0.001$)."

Reference

1. Chen QY, Xie JW, Zhong Q, et al. Safety and Efficacy of Indocyanine Green Tracer-Guided Lymph Node Dissection During Laparoscopic Radical Gastrectomy in Patients With Gastric Cancer: A Randomized Clinical Trial. *JAMA Surg.* 2020 Apr 1;155(4):300-311.

3. In your limitations you mention that proximal gastrectomies are more common in Western practice, which may not be supported in practice. Do you mean total gastrectomies?

The author's answer: Thank you for your comments. To avoid any misunderstanding, we have modified the statement to "Third, the study was conducted in Eastern Asia, so it is not clear whether the results could be generalized to Western settings"

4. A major concern / barrier for interpreting this study results are not addressed in your discussion: that in the western nations most patients with locally advanced resectable gastric cancer receive neoadjuvant therapy. The question remains whether or not these results are reproducible after receiving neoadjuvant chemotherapy.

The author's answer: Thank you for your valuable comment. With the reporting of various large-scale prospective clinical studies¹⁻⁴, multiple guidelines including the *Japanese Gastric Cancer Treatment Guidelines* recommend neoadjuvant chemotherapy combined with surgical radical resection, rather than direct resection for GC patients with Bulky N^{1,5}. Obviously, neoadjuvant chemotherapy will affect the prognosis and subsequent treatment decisions of GC patients⁶⁻⁷, leading to significant heterogeneity in the study population. Therefore, we excluded patients with enlarged or bulky regional LNs with a diameter of more than 3 cm in our study. Additionally, we will conduct ICG-related research on patients with LN enlargement to explore the role of ICG fluorescence navigation technology in LN dissection for this type of patient. For instance, our center is currently conducting an RCT study of ICG for GC receiving neoadjuvant chemotherapy (NCT04611997), and we look forward to future reports on the application of ICG in patients who received neoadjuvant chemotherapy. We have added these contents to the *Discussion* in the revised manuscript.

Reference

1. Japanese Gastric Cancer Association. Japanese Gastric Cancer Treatment Guidelines 2021 (6th edition). Gastric Cancer. 2023 Jan;26(1):1-25.
2. Iwasaki Y, Terashima M, Mizusawa J, et al. Gastrectomy with or without neoadjuvant S-1 plus cisplatin for type 4 or large type 3 gastric cancer (JCOG0501): an open-label, phase 3, randomized controlled trial. Gastric Cancer. 2021 Mar;24(2):492-502.
3. Kang YK, Yook JH, Park YK, et al PRODIGY: a phase III study of neoadjuvant docetaxel, oxaliplatin, and S-1 plus surgery and adjuvant S-1 versus surgery and adjuvant S-1 for resectable advanced gastric cancer. J Clin Oncol. 2021;39:2903-13.
4. Zhang X, Liang H, Li Z, et al Perioperative or postoperative adjuvant oxaliplatin with S-1 versus adjuvant oxaliplatin with capecitabine in patients with locally advanced gastric or gastroesophageal junction adenocarcinoma undergoing D2 gastrectomy (RESOLVE): an open-label, superiority and non-inferiority, phase 3 randomised controlled trial. Lancet Oncol. 2021;22(8):1081–92.
5. Ajani JA, et al. Gastric Cancer, Version 2.2022, NCCN Clinical Practice Guidelines in Oncology. Journal of the National Comprehensive Cancer Network : JNCCN 20, 167-192 (2022).
6. Katayama H, Tsuburaya A, Mizusawa J, et al. An integrated analysis of two phase II trials (JCOG0001 and JCOG0405) of preoperative chemotherapy followed by D3 gastrectomy for gastric cancer with extensive lymph node metastasis. Gastric Cancer. 2019 Nov;22(6):1301-1307.
7. Takahari D, Ito S, Mizusawa J, et al; Stomach Cancer Study Group of the Japan Clinical Oncology Group. Long-term outcomes of preoperative docetaxel with cisplatin plus S-1 therapy for gastric cancer with extensive nodal metastasis (JCOG1002). Gastric Cancer. 2020 Mar;23(2):293-299.

Reviewers' Comments:

Reviewer #2:

Remarks to the Author:

Following the response to question 1 and 2 of reviewer 4 there's one follow-up question that will have to be addressed: With ICG the lymph node yield is significantly higher and consequently these patients do better. How to the authors explain this? Is it simply bc more nodes are harvested, not necessarily leading to more cancer resected?

Reviewer #3:

Remarks to the Author:

Regarding my previous comment on univariable vs multivariable: Please include a statement that the competing risk regression was not adjusted for any confounders along with justification. However, for the competing risk regression for any recurrence, it seems the numbers are sufficient to adjust for confounders and a pvalue adjusted for confounders would be recommended. Similarly, pvalues for interactions are provided but it is not clear if these models were adjusted for the relevant confounders. The concern still remains that some results are adjusted for confounders while others are unadjusted; this not only adds confusion, it also raises the question whether some of the significant findings hold after adjustment for confounders.

RESPONSE TO REVIEWERS

Title: “**Long-Term Outcomes of Indocyanine Green Fluorescence Imaging-Guided versus Conventional Laparoscopic Lymphadenectomy for Gastric Cancer**” We are extremely grateful to the editor and the anonymous reviewers for their valuable comments and suggestions, which have helped us to improve the quality of the manuscript. We thank the reviewers and editors for taking the time to review our manuscript and appreciate your comments on this paper.

After careful consideration of the reviewers' comments, we have revised the manuscript accordingly. Our responses to the comments are as follows:

REVIEWER COMMENTS

Reviewer #2 (Remarks to the Author):

1. Following the response to question 1 and 2 of reviewer 4 there's one follow-up question that will have to be addressed: With ICG the lymph node yield is significantly higher and consequently these patients do better. How to the authors explain this? Is it simply bc more nodes are harvested, not necessarily leading to more cancer resected?

Response: We thank the reviewer for raising this point. The possible reasons for the survival benefits of ICG-guided surgery for GC patients may be as follows: (i) First, this study is a rigorous prospective controlled trial, in which 258 patients who underwent GC radical surgery (doi:10.1001/jamasurg.2019.6033)¹ were included after random grouping. We found that the frequency of tumor burden factors, such as tumor stage and size, were similar between the two groups of patients, and both groups had negative surgical margins in postoperative pathological reports, thereby, indicating that the primary lesions had been effectively eradicated. (ii) Second, the results of this study have shown that within the specified scope of dissection, greater the number of LNs retrieved, better was the long-term survival of GC patients, thus

corroborating with previous studies²⁻⁶. Therefore, for patients with GC, especially those with locally advanced GC, complete dissection of metastatic LNs and a reduction in missed dissection of metastatic LNs are of great significance for accurate staging and subsequent treatment options (Line 229-233). Furthermore, the more LNs are removed, the more positive and negative LNs with possible micrometastasis will increase.. Dissecting a sufficient number of LNs in the standard lymphadenectomy area is necessary for accurate disease staging and to avoid missed dissection of metastatic LNs⁷, thus having a positive impact on patient prognosis (Line 254-258).

Furthermore, during surgery for GC, particularly in locally advanced GC, improper manipulation of the lymphatic adipose tissue often leads to the release of free cancer cells from the lymphovascular pedicle and metastatic LN, thereby increasing the risk of recurrence⁸⁻¹¹. Given that ICG can track the lymphatics and LNs well under high-resolution laparoscopic imaging, it may reduce the dissemination of free cancer cells caused by improper surgical procedures, such as incorrect handling of LN-containing tissue by surgeons to a certain extent (Line 283-290). The above evidence suggests that the improvement in prognosis relies on greater number of nodes being harvested, which does not necessarily lead to more cancer resection. We look forward to future multicenter studies (CLASS-11 trial: NCT04593615) providing higher-level evidence in the field of ICG-guided surgery for survival benefits in patients with gastric cancer. The above changes have been supplemented and marked in red in the Discussion section (Line 229-233, Line 254-258, Line 283-290).

Reference

1. Chen QY, Xie JW, Zhong Q, et al. Safety and Efficacy of Indocyanine Green Tracer-Guided Lymph Node Dissection During Laparoscopic Radical Gastrectomy in Patients With Gastric Cancer: A Randomized Clinical Trial. *JAMA Surg.* 2020 Apr 1;155(4):300-311.
2. Smith DD, Schwarz RR, Schwarz RE. Impact of total lymph node count on staging and survival after gastrectomy for gastric cancer: data from a large US-population database. *Journal of clinical oncology: official journal of the American Society of Clinical Oncology*

- 23, 7114-7124 (2005).
3. Li GZ, Doherty GM, Wang J. Surgical Management of Gastric Cancer: A Review. *JAMA Surg.* 2022 May 1;157(5):446-454.
 4. Gholami S, *et al.* Number of Lymph Nodes Removed and Survival after Gastric Cancer Resection: An Analysis from the US Gastric Cancer Collaborative. *J Am Coll Surg.* 2015 Aug;221(2):291-9.
 5. Smyth EC, *et al.* ESMO Guidelines Committee. Gastric cancer: ESMO Clinical Practice Guidelines for diagnosis, treatment, and follow-up. *Ann Oncol.* 2016;27(suppl 5):v38-v49.
 6. Datta J, *et al.* Implications of Lymph Node Staging on Selection of Adjuvant Therapy for Gastric Cancer in the United States: A Propensity Score-matched Analysis. *Ann Surg.* 2016 Feb;263(2):298-305.
 7. Kim TH, *et al.* Assessment of the Completeness of Lymph Node Dissection Using Near-infrared Imaging with Indocyanine Green in Laparoscopic Gastrectomy for Gastric Cancer. *Journal of gastric cancer* **18**, 161-171 (2018).
 8. Takebayashi K, *et al.* Surgery-induced peritoneal cancer cells in patients who have undergone curative gastrectomy for gastric cancer. *Annals of surgical oncology* **21**, 1991-1997 (2014).
 9. Han TS, *et al.* Dissemination of free cancer cells from the gastric lumen and from perigastric lymphovascular pedicles during radical gastric cancer surgery. *Annals of surgical oncology* **18**, 2818-2825 (2011).
 10. Marutsuka T, *et al.* Mechanisms of peritoneal metastasis after operation for non-serosa-invasive gastric carcinoma: an ultrarapid detection system for intraperitoneal free cancer cells and a prophylactic strategy for peritoneal metastasis. *Clin Cancer Res.* 2003 Feb;9(2):678-85.
 11. Ren K, *et al.* Development of the Peritoneal Metastasis: A Review of Back-Grounds, Mechanisms, Treatments and Prospects. *J Clin Med.* 2022 Dec 23;12(1):103.

Reviewer #3 (Remarks to the Author):

1. Regarding my previous comment on univariable vs. multivariable: Please include a statement that the competing risk regression was not adjusted for any confounders along with justification. However, for the competing risk regression for any recurrence, it seems the numbers are sufficient to adjust for confounders and a pvalue adjusted for confounders would be **recommended**. Similarly, pvalues for interactions are provided but it is not clear if these models were adjusted for the relevant confounders. The concern still remains that some results are adjusted for confounders while others are unadjusted;

this not only adds confusion, it also raises the question whether some of the significant findings hold after adjustment for confounders.

Response: We thank the reviewer for this valuable comment. The reviewer's request to adjust for confounding factors is reasonable and highly important. Following your instructions, we have adjusted for confounding factors (sex, AJCC7th staging, and adjuvant chemotherapy) in the competing risk model and added the HR (95%) and corresponding *P* values adjusted for confounders in the recurrence analysis in **Table 2**. The results (Lines 121, 146-147, 149, 214-220) are consistent with previous findings. Additionally, we have provided supplemental information in **Supplemental Digital Content 3** and **Table 2**, stating, "Multivariable Cox regression was used for all-cause death, after adjustment for sex, AJCC7th staging and adjuvant chemotherapy."

Furthermore, as per the reviewer's suggestion, we have adjusted for confounding factors (Sex, AJCC7th staging, and adjuvant chemotherapy) in the results of the interaction *P*-test based on the approach used in previous studies¹⁻⁵. The results are consistent with the previous findings and are as follows: "Among the full cohort, there are no significant interactive effect between ICG and LN dissection compliance on OS and DFS (*P*-interaction for OS= 0.077, adjusted *P*-interaction for OS= 0.061; *P*-interaction for DFS= 0.125, adjusted *P*-interaction for DFS= 0.094). Among patients with pN+ stage disease, there was a significant interactive effect of ICG and LN dissection noncompliance on OS and DFS (*P*-interaction for OS= 0.033, adjusted *P*-interaction for OS= 0.028; *P*-interaction for DFS= 0.039, adjusted *P*-interaction for DFS= 0.033; **eTable 6**"). We have made these changes in the revised manuscript's **Results** section (**Lines 214-220**).

Based on the aforementioned findings, we observed that even after adjusting for confounding factors, our main results remained consistent with those of the earlier version of the manuscript. In comparison to conventional lymphadenectomy, ICG-guided laparoscopic lymphadenectomy is more effective in prolonging survival in patients with resectable GC. Finally, we

eagerly await the emergence of large-scale studies (CLASS-11 trial: NCT04593615) to validate our findings and provide a higher level of confirmation in the field of evidence-based medicine. We have made supplementary revisions to the **Method** section (Line 459-462), **Results** (Lines 121, 146-147, 149, 214-220), and **Supplemental Digital Content 3**.

Reference

1. Knol MJ, VanderWeele TJ, Groenwold RH, *et al.* Rovers MM, Grobbee DE. Estimating measures of interaction on an additive scale for preventive exposures. *Eur J Epidemiol.* 2011 Jun;26(6):433-8.
2. Rod NH, Lange T, Andersen I, *et al.* Additive interaction in survival analysis: use of the additive hazards model. *Epidemiology.* 2012 Sep;23(5):733-7.
3. Li R, Chambless L. Test for additive interaction in proportional hazards models. *Ann Epidemiol.* 2007 Mar;17(3):227-36.
4. Xu RB, Kong X, Xu BP, *et al.* Longitudinal association between fasting blood glucose concentrations and first stroke in hypertensive adults in China: effect of folic acid intervention. *Am J Clin Nutr.* 2017 Mar;105(3):564-570.
5. Wesselink E, Kok DE, Bours MJL, *et al.* Vitamin D, magnesium, calcium, and their interaction in relation to colorectal cancer recurrence and all-cause mortality. *Am J Clin Nutr.* 2020 May 1;111(5):1007-1017.

Table 2. Frequencies of Causes of First Recurrence and Death Within 3 Years After Surgery in ICG and Non-ICG Groups

Events	Surgery, No. (%)		Risk Difference ^a	Hazard Ratio (95% CI) ^b	P Value ^c	Adjusted Hazard Ratio (95% CI) ^d	Adjusted P Value ^e
	ICG Group (n = 129)	Non-ICG Group (n = 129)					
Any recurrence^f	23 (17.8)	40 (31.0)	-0.131	0.53 (0.32-0.89)	0.017	0.54 (0.32-0.91)	0.020
Local	2 (1.6)	10 (7.8)	-0.073	0.19 (0.04-0.85)	0.030	0.22 (0.05-0.99)	0.048
Peritoneum	9 (7.0)	10 (7.8)	-0.009	0.87 (0.35-2.13)	0.752	0.96 (0.39-2.36)	0.923
Liver	2 (1.6)	6 (4.7)	-0.032	0.33 (0.07-1.62)	0.170	0.31 (0.06-1.56)	0.155
Multiple sites ^g	4 (3.1)	5(3.9)	-0.009	0.79 (0.21-2.92)	0.718	0.93 (0.25-3.51)	0.917
Other or uncertain sites ^h	6 (4.7)	9 (7.0)	-0.026	0.64 (0.23-1.81)	0.402	0.55 (0.19-1.60)	0.274
Cause of deathⁱ	18 (14.0)	34 (26.4)	-0.124	0.50 (0.28-0.89)	0.018	0.54 (0.30-0.96)	0.035
Gastric cancer	17 (94.4)	32 (94.1)	-0.116	0.50 (0.28-0.91)	0.022	0.53 (0.29-0.96)	0.037
Other causes ^j	1 (5.6)	2 (5.9)	-0.011	0.47 (0.04-5.21)	0.540	0.48 (0.04-5.43)	0.556

^a Except for all-cause death, the risk difference was calculated by subtracting the cumulative incidence in the first 3 years of the Non-ICG group from that of the ICG group, in presence of

competing events; for all-cause death, the risk difference was calculated by subtracting the 3-year overall survival rate of the Non-ICG group from that of the ICG group.

^b Except for all-cause death, competing-risks survival regression was used to derive the hazard ratio, 95% CI, and P value. For total recurrence, all-cause death was the competing event; for the specific types of recurrence, other types of recurrence and death were the competing events; for gastric cancer cause of death, other causes of death were the competing events, and vice versa. **Univariable Cox regression was used for all-cause death.** Non-ICG group is the reference group.

^c P value for the hazard ratios.

^d **Multivariable Cox regression was used for all-cause death, adjustment for sex, AJCC7th staging and adjuvant chemotherapy.**

^e **Adjusted P** value for the hazard ratios.

^f Refers only to first-time recurrence, even though patients can have recurrence at multiple times.

^g Includes patients who have recurrence simultaneously in 2 or more metastatic sites, including peritoneum, liver, lung, bone, brain, distant lymph node, or other hematogenous metastatic sites.

^h Includes hematogenous recurrence at sites other than liver (ie, lung, bone, brain, adrenal gland), recurrence at distant lymph node, and recurrence at uncertain sites.

ⁱ Post hoc exploratory outcomes.

^j Includes other cancers, diseases other than cancer, unintentional injuries, and unknown causes.

eTable 6. Interaction of ICG with Lymph Node Dissection Noncompliance in Relation to Overall Survival and Disease-Free Survival*

Model	Characteristic	All Patients				pN+ Patients			
		Overall Survival		Disease-Free Survival		Overall Survival		Disease-Free Survival	
		HR (95%CI)	P-value	HR (95%CI)	P-value	HR (95%CI)	P-value	HR (95%CI)	P-value
Groups									
Model 1[#]	Non-ICG	1 [Reference]	-	1 [Reference]	-	1 [Reference]	-	1 [Reference]	-
	ICG	0.50 (0.28-0.89)	0.018	0.53 (0.32-0.88)	0.014	0.52 (0.29-0.93)	0.027	0.53 (0.32-0.88)	0.015
Model 2[†]	Lymph Nodes Dissection Noncompliance								
	Noncompliance	1 [Reference]	-	1 [Reference]	-	1 [Reference]	-	1 [Reference]	-
	Compliance	0.80 (0.46-1.37)	0.415	0.75 (0.46-1.21)	0.241	0.67 (0.38-1.17)	0.156	0.61 (0.37-0.99)	0.047
Model 3^{††}	P for interaction		0.077		0.125		0.033		0.039
Model 4[‡]	Adjusted P for interaction		0.061		0.094		0.028		0.033

* Analyzed using the Cox proportional hazards model.

[#] Univariable Cox regression analysis results of the ICG and non-ICG groups on overall survival and disease-free survival.

[†] Univariable Cox regression analysis results of lymph node dissection noncompliance and compliance on overall survival and disease-free survival.

†† The multiplicative interactive relationship of ICG and lymph node dissection compliance with overall and disease-free survival.

‡ The multiplicative interactive relationship was adjusted for sex, AJCC7th staging and adjuvant chemotherapy.

Abbreviations: GC, gastric cancer; ICG, indocyanine green.

Reviewers' Comments:

Reviewer #2:

None

Reviewer #3:

Remarks to the Author:

Please clarify the entries in the legend of Table 2 referring to "all-cause death" which is not separately shown in the table.

No further comments.

Nature communications

RESPONSE TO REVIEWERS

Title: "Indocyanine Green Fluorescence Imaging-Guided versus Conventional Laparoscopic Lymphadenectomy for Gastric Cancer: Long-term Outcomes of a Phase 3 Randomised Clinical Trial" We are extremely grateful to the editor and the anonymous reviewers for their valuable comments and suggestions, which have helped us to improve the quality of the manuscript. We thank the reviewers and editors for taking the time to review our manuscript and appreciate your comments on this paper.

After careful consideration of the reviewers' comments, we have revised the manuscript accordingly. Our responses to the comments are as follows:

REVIEWERS' COMMENTS

Reviewer #3 (Remarks to the Author):

Please clarify the entries in the legend of Table 2 referring to "all-cause death" which is not separately shown in the table.

No further comments.

Response: We thank the reviewer for raising this point. To facilitate understanding by reviewers and readers, we have modified "cause of death" to "all-cause death" in the Table 2, and "Any recurrence" to "Recurrence" in Table 2, supplemented "All-cause death includes death from gastric cancer and other causes" in the footnote of Table 2, and made corresponding changes in the Method of revised manuscript and Supplementary Information 3.

Table 2. Frequencies of Causes of First Recurrence and Death Within 3 Years After Surgery in ICG and Non-ICG Groups

Events	Surgery, No. (%)		Risk Difference ^a	Hazard Ratio (95% CI) ^b	P Value ^c	Adjusted Hazard Ratio (95% CI) ^d	Adjusted P Value ^e
	ICG Group (n = 129)	Non-ICG Group (n = 129)					
Recurrence ^f	23 (17.8)	40 (31.0)	-0.131	0.53 (0.32-0.89)	0.017	0.54 (0.32-0.91)	0.020

Local	2 (1.6)	10 (7.8)	-0.073	0.19 (0.04-0.85)	0.030	0.22 (0.05-0.99)	0.048
Peritoneum	9 (7.0)	10 (7.8)	-0.009	0.87 (0.35-2.13)	0.752	0.96 (0.39-2.36)	0.923
Liver	2 (1.6)	6 (4.7)	-0.032	0.33 (0.07-1.62)	0.170	0.31 (0.06-1.56)	0.155
Multiple sites ^g	4 (3.1)	5(3.9)	-0.009	0.79 (0.21-2.92)	0.718	0.93 (0.25-3.51)	0.917
Other or uncertain sites ^h	6 (4.7)	9 (7.0)	-0.026	0.64 (0.23-1.81)	0.402	0.55 (0.19-1.60)	0.274
All-cause deathⁱ	18 (14.0)	34 (26.4)	-0.124	0.50 (0.28-0.89)	0.018	0.54 (0.30-0.96)	0.035
Gastric cancer	17 (94.4)	32 (94.1)	-0.116	0.50 (0.28-0.91)	0.022	0.53 (0.29-0.96)	0.037
Other causes ^j	1 (5.6)	2 (5.9)	-0.011	0.47 (0.04-5.21)	0.540	0.48 (0.04-5.43)	0.556

^a For recurrence, the risk difference was calculated by subtracting the cumulative incidence in the first 3 years of the Non-ICG group from that of the ICG group, in presence of competing events; for all-cause death, the risk difference was calculated by subtracting the 3-year overall survival rate of the Non-ICG group from that of the ICG group.

^b For recurrence, competing-risks survival regression was used to derive the hazard ratio, 95% CI, and *P* value. For total recurrence, all-cause death was the competing event; for the specific types of recurrence, other types of recurrence and death were the competing events; for gastric cancer cause of death, other causes of death were the competing events, and vice versa. Univariable Cox regression was used for recurrence and all-cause death. Non-ICG group is the reference group.

^c *P* value for the hazard ratios.

^d Multivariable Cox regression was used for recurrence and all-cause death, adjustment for sex, AJCC7th staging and adjuvant chemotherapy.

^e Adjusted *P* value for the hazard ratios.

^f Refers only to first-time recurrence, even though patients can have recurrence at multiple times.

^g Includes patients who have recurrence simultaneously in 2 or more metastatic sites, including peritoneum, liver, lung, bone, brain, distant lymph node, or other hematogenous metastatic sites.

^h Includes hematogenous recurrence at sites other than liver (ie, lung, bone, brain, adrenal gland), recurrence at distant lymph node, and recurrence at uncertain sites.

ⁱ Post hoc exploratory outcomes. All-cause death includes death from gastric cancer and other causes.

^j Includes other cancers, diseases other than cancer, unintentional injuries, and unknown causes.

To The Editors

1. In accordance with the journal's requirements, we have added the study's purpose in the article abstract, as follows:

Abstract

Indocyanine green (ICG) fluorescence imaging-guided lymphadenectomy has

been proven effective in increasing the number of lymph nodes (LNs) retrieved in laparoscopic gastrectomy for gastric cancer (GC). We previously reported the short-term efficacy of ICG imaging in laparoscopic gastrectomy. This study reports its unclear long-term oncological efficacy. In this phase 3, open-label, randomized clinical trial (NCT03050879), 266 eligible patients with potentially resectable GC are randomly (1:1 ratio) assigned to either the ICG or non-ICG group. The primary outcome is the number of LNs retrieved. The secondary outcomes is firstly reported, including three-year overall survival (OS), three-year disease-free survival (DFS), and recurrence patterns. The per-protocol analysis set population is used for all analyses (258 patients, ICG [n=129] vs. non-ICG group [n=129]). The mean total LNs retrieved in the ICG group significantly exceeds that in the non-ICG group (50.5 ± 15.9 vs 42.0 ± 10.3 , $P < 0.001$). Both OS and DFS in the ICG group are significantly better than that in the non-ICG group (log-rank $P = 0.015$; log-rank $P = 0.012$, respectively). There is a difference in the overall recurrence rates between the ICG and non-ICG groups (17.8% vs 31.0%). Compared with conventional lymphadenectomy, ICG guided laparoscopic lymphadenectomy is safe and effective in prolonging survival among patients with resectable GC.

We would be very appreciative of your kind consideration of this paper.

Sincerely,

Best regards,

Chang-Ming Huang, MD, Professor

Department of Gastric Surgery, Fujian Medical University Union Hospital,

E-mail: hcmlr2002@163.com